# Net ecosystem carbon exchange of a dry temperate eucalypt forest

**Nina Hinko-Najera**[1], Peter Isaac[2], Jason Beringer[3], Eva van Gorsel[4], Cacilia Ewenz[5], Ian McHugh[6], Jean-François Exbrayat[7], Stephen J. Livesley[8], Stefan K. Arndt[8]

[1]School of Ecosystem and Forest Sciences, The University of Melbourne, 4 Water Street, Creswick, VIC 3363, Australia

[2] CSIRO Marine and Atmospheric Research, Pye Laboratory, Clunies Ross Street, Acton, ACT 2600, Australia

[3]School of Earth and Environment, The University of Western Australia, Crawley, WA, 6009, Australia

[4]Fenner School of Environment and Society, The Australian National University, ACT, Canberra, Australia

[5]Airborne Research Australia, Flinders University, Salisbury South, SA, 5106, Australia

[6]School of Earth, Atmosphere and Environment, Monash University, Clayton, VIC 3800, Australia

[7]School of GeoSciences and National Centre for Earth Observation, University of Edinburgh, Edinburgh, EH9 3FF, UK

[8]School of Ecosystem and Forest Sciences, The University of Melbourne, 500 Yarra Boulevard, Richmond, VIC 3121, Australia

*Correspondence to*: Nina Hinko-Najera (e-mail: n.hinko.najera@gmail.com)

**Abstract.** Forest ecosystems play a crucial role in the global carbon cycle by sequestering a considerable fraction of anthropogenic $CO_2$ thereby contributing to climate change mitigation. However, there is a gap in our understanding about the carbon dynamics of eucalypt (broadleaf evergreen) forests in temperate climates, which might differ from temperate evergreen coniferous or deciduous broadleaved forests given their fundamental differences in physiology, phenology and growth dynamics. To address this gap we undertook a three year study (2010 – 2012) of eddy covariance measurements in a dry temperate eucalypt forest in south-eastern Australia. We determined the annual net carbon balance and investigated the temporal (seasonal and inter-annual) variability and environmental controls of net ecosystem carbon exchange (NEE), gross primary productivity (GPP) and ecosystem respiration (ER). The forest was a large and constant carbon sink throughout the study period, even in winter, with an overall mean NEE of -1234 ± 109 (SE) g C m$^{-2}$ yr$^{-1}$. Estimated annual ER was similar for 2010 and 2011 but decreased in 2012 ranging from 1603 to 1346 g C m$^{-2}$ yr$^{-1}$ whereas GPP annual showed no significant inter-annual variability with an mean annual estimate of 2728 ± 39 g C m$^{-2}$ yr$^{-1}$. All ecosystem carbon fluxes had a pronounced seasonality with GPP being greatest during spring and summer and ER during summer whereas peaks of NEE

occurred in early spring and again in summer. High NEE in spring was likely caused by a delayed increase in ER due to low temperatures. A strong seasonal pattern in environmental controls of day time and night time NEE was revealed. Day time NEE was equally explained by incoming solar radiation and air temperature whereas air temperature was the main

environmental driver of night time NEE. The forest experienced unusual above average annual rainfall during the first two years of this three year period so that soil water content remained relatively high and the forest was not water limited. Our results show the potential of temperate eucalypt forests to sequester large amounts of carbon when not water limited. However, further studies using bottom-up approaches are needed to validate measurements from EC flux tower and to account for a possible underestimation in ER due to advection fluxes.

**Keywords.** net ecosystem productivity, south-eastern Australia, 2011 La Niña, OzFlux, Wombat State Forest, dry sclerophyll forest, random forest approach

# 1 Introduction

Terrestrial ecosystems, together with the ocean, take up more than half of the yearly anthropogenic $CO_2$ emissions and their combined sink strength has increased over the past five decades in step with increased emissions (Ballantyne et al., 2012; Le Quere et al., 2013; 2015). The terrestrial sink has been mostly attributed to the world's forest ecosystems over the last two decades (Le Quere et al., 2013; Pan et al., 2011) and only recently the importance of semi-arid ecosystems in the global carbon sink has been identified (Ahlström et al., 2015; Poulter et al., 2014). Even so, forests play a crucial role in the global

carbon cycle and climate change mitigation (IPCC, 2013; Pan et al., 2011).

Nonetheless, uncertainty remains regarding the future trend and strength of this terrestrial carbon sink (Ciais et al., 2013; Mystakidis et al., 2016; Reichstein et al., 2013; Sitch et al., 2015). This is mainly related to the high inter-annual variability in the carbon uptake of ecosystems because of regional and even global variations in climate year-to-year (Ahlström et al., 2015; Reichstein et al., 2013). The balance between gross primary productivity (GPP) and ecosystem respiration (ER) – the

net ecosystem carbon exchange NEE) and can be positive (a carbon sink) or negative (a carbon source), although other carbon exchanges such as dissolved organic transport and/or disturbance provide a true net ecosystem carbon balance (NECB) (Chapin et al., 2006). Hence, variability in NEE is dependent on variations of the component fluxes GPP and ER and their responses to climate and resource availability (Ahlström et al., 2015; Ciais et al., 2013; Reichstein et al., 2013). While some studies primarily attribute inter-annual variability in NEE to changes in respiration (Cox et al., 2000; Valentini

et al., 2000), others point to a primary dependence on the variability in ecosystem GPP (Ahlström et al., 2015; Jung et al., 2011; Sitch et al., 2015).

Across various ecosystems the main environmental factors controlling GPP have been identified being solar radiation, water vapour pressure deficit (VPD) and leaf area index (LAI), whereas temperature and soil moisture are the main environmental drivers of ER (Baldocchi, 2008; Beringer et al., 2016a; Yi et al., 2010). Variability in NEE has also been demonstrated to be

strongly influenced by variation in water availability (i.e. changes in rainfall). For instance, the overall effect of drought conditions has been shown to decrease NEE but it often varies which component, GPP or ER, drought conditions have the greatest impact (Ciais et al., 2005; Reichstein et al., 2007; Schlesinger et al., 2015; Zhao and Running, 2010). It is therefore critical to assess the carbon balance of ecosystems to improve our knowledge of processes controlling NEE and their

response to variability of environmental drivers and climate change. Another factor contributing to the uncertainty of future

terrestrial carbon sinks is the still limited empirical data available on forest carbon dynamics to better constrain uncertainties of global and continental process-based carbon models and/or to improve data-driven model frameworks (Haverd et al., 2013a; Jung et al., 2011; Keenan et al., 2012; Roxburgh et al., 2004).

Forests in Australia occupy around 19% of the continent and account for about 3% to forested area worldwide (ABARES, 2013) and until recently their potential contribution to the global carbon cycle has not been considered. The role of

Australian ecosystems generally in the global carbon cycle has had recent attention in the light of the 2011 strong La Niña event and global record terrestrial carbon sink, where Australian ecosystems, particularly semi-arid ecosystems, played a major role in the continental and global carbon uptake anomaly (Haverd et al., 2013b; 2016; Poulter et al., 2014). Although semi-arid ecosystems have been suggested as dominant drivers in inter-annual variability and trends of the global net carbon sink (Ahlström et al., 2015), little is known about how Australian temperate eucalypt (broadleaved evergreen) forests may

contribute to the global sink and inter-annual variability. Two thirds of native forests in Australia are eucalypt forests (92 M ha) and dry temperate eucalypt forests account for the largest proportion (37% or 8.3 M ha) of forest ecosystems in south-eastern and south-western Australia and are of high socio-economic value (ABARES, 2013). Growth and regeneration of temperate forests in the Northern Hemisphere are considered to account for the increasing global terrestrial carbon sink (Pan et al., 2011), although a recent study showed a decline in this trend (Sitch et al., 2015). While studies of the carbon balance

in the Northern Hemisphere temperate forests are abundant, there are only a handful of studies that have been undertaken in temperate eucalypt forests in Australia and none of these in dry temperate eucalypt forests (Beringer et al., 2016a; Keith et al., 2009; 2012; Kilinc et al., 2012; 2013; Leuning et al., 2005; van Gorsel et al., 2013). The behaviour of temperate deciduous broadleaved or evergreen coniferous forests in the Northern Hemisphere cannot be presumed to be an analogue for temperate eucalypt forests. Apart from being broadleaf evergreen, with mostly sclerophyllous leaves, a key trait of

eucalypt forests in Australia is the ability to rapidly and opportunistically respond to changing, either favourable or stressful, environmental conditions (Jacobs, 1955; Keith, 1997). This is an adaptation to disturbances such as fire or drought that are a major component of ecosystems on the Australian continent (ABARES, 2013; Beringer et al., 2015; Whitehead and Beadle, 2004). Moreover, Australian forests are generally water and nutrient limited and soils are highly weathered (Attiwill and

Adams, 1993; Whitehead and Beadle, 2004). Keith et al. (2009) showed that a wet temperate eucalypt forest had a high-carbon uptake capacity compared with other forests globally when not limited by water availability. No studies have been published on ecosystem carbon exchange in dry temperate eucalypt forests, where rainfall is considerably lower and soil moisture likely to be a greater limiting factor.

The aim of the study was to assess the carbon uptake potential of a dry temperate eucalypt forest and to gain an understanding of its temporal carbon exchange dynamics and controls thereof by using the eddy covariance (EC) technique (Baldocchi, 2008; 2003; Hutley et al., 2005) as part of the regional OzFlux network (Beringer et al., 2016a).

Therefore the objectives of our study were to 1) investigate seasonal and inter-annual variability in net ecosystem carbon exchange (NEE), gross primary productivity (GPP) and ecosystem respiration (ER), 2) identify the environmental controls of these $CO_2$ ecosystem fluxes on seasonal and inter-annual time scales, and 3) quantify annual estimates of NEE and its component fluxes in a dry temperate eucalypt forest.

## 2 Materials and methods

### 2.1 Site description

The Wombat State Forest OzFlux tower site (Fluxnet ID: AUS-Wom) is located in the Wombat State Forest, Victoria, about 120 km west of Melbourne, Australia (37° 25' 20.5" S, 144° 05' 39.1" E). The Wombat State Forest is classified as dry sclerophyll eucalypt forest or Open (crown cover >50-80%) forest (ABARES, 2013) is dominated by three broadleaved evergreen tree species: *Eucalyptus obliqua* (L'Hérit.), *Eucalyptus rubida* (Deane & Maiden) and *Eucalyptus radiata* (Sieber ex DC). General forest history includes harvesting and patchy occurrences of bushfires. Selective harvesting occurred until early 1970 when replaced by a more intensive shelterwood (two-stage clear-felling) system (Poynter, 2005). Since 2003 the Wombat State Forest has been under community forest management, a cooperative between state government and local community (Poynter, 2005) and harvesting has been strongly reduced. Forest management practices also include periodic low fire intensity prescribed fires and firewood collection in designated areas.

The study site is a secondary regrowth forest (DSE, 2012), of mixed age, with an average canopy height of 22 m (Griebel et al., 2015), a basal area of 37 $m^2$ $ha^{-1}$ (Moore, 2011) and a LAI of 1.8 (Griebel et al., 2016; Moore, 2011).The area was

selectively harvested last in the early 1970s with the last bushfire on the outskirts of the study site recorded in 1982 and no recorded history of prescribed fires. The flux tower is located on a ridge at a mean altitude of 706 m a.s.l. and the terrain

within the footprint is relatively level to the east of the tower and with gently sloping gullies (<8∘) towards the southwest and northwest (Griebel et al., 2016). The understorey is sparse and dominated by Austral bracken (*Pteridium esculentum* (G. Forst.) Cockayne), Forest wire-grass (*Tetrarrhena juncea* R. Br.), Tussock Grass (*Poa sieberiana* Sprengel), herbs (e.g. *Gonocarpus tetragynus* Labill., *Viola hederacea* Labill.) and rushes (*Lomandra* spp.) (Tolhurst, 2003). The climate is cool temperate to Mediterranean with wet, cold winters and dry, warm/hot summers. Long-term (2001-2013) mean annual air

temperature was $12.1 \pm 0.1$ $^{\circ}$C with mean monthly maximum air temperatures of $26.3 \pm 0.5$ $^{\circ}$C in January and mean minimum air temperatures of $3.2 \pm 0.1$ $^{\circ}$C in July (nearest Bureau of Meteorology (BOM) station Ballarat, 28km SW, Fig. 1a). The silty clay soil overlying clay derived from Ordovician marine sediments and are classified as Acidic-mottled, Dystrophic, Yellow Dermosol (Robinson et al., 2003), moderate to highly weathered and exhibit low fertility. The long-term (1901-2014) mean annual rainfall at the nearest rainfall BOM station (Daylesford, 11km N, Fig. 1b) is $879 \pm 18$ mm with the

highest rainfall occurring during winter and spring. For overview and more detailed site characteristics see Table 1.

## 2.2 Instrumentation and data aquisition

The guyed Eddy – Covariance (EC) flux tower was established in January 2010 within a fenced compound. The micro-meteorological measurement system was installed at 30 m height and consisted of an open-path infrared gas analyser (IRGA,

Li-7500, LI-COR, Lincoln, USA) that measures $CO_2$ and water vapour concentrations and atmospheric pressure, and a 3D – sonic anemometer (CSAT3, Campbell Scientific Inc., Logan, USA) that measures turbulent wind vectors and virtual air temperature. Instantaneous measurements were carried out at 10 Hz and stored on a CF-card. Furthermore calculated covariances with a 30 min averaging period were stored on a data logger (CR-3000, Campbell Scientific Inc., Logan, USA). Prior to the calculation of covariances at the end of a 30 min averaging period, 10 Hz data were filtered by the data logger

depending on diagnostic information from both the sonic anemometer and IRGA in which data spikes got removed (Isaac et al., 2016). Concurrent measurements of environmental variables included: air temperature (Ta) and absolute and relative humidity (HMP-45C probe, Vaisala, FIN) at 2 m and 30 m height, incoming and reflected shortwave radiation and

atmospheric and surface emitted longwave radiation with a CNR1 net radiometer ( Kipp and Zonen, Delft, NLD) at 30 m height, rainfall with a tipping bucket rain gauge (CS702, Hydrological Services Pty Ltd., Sydney, AUS) at 1 m height, soil
heat flux, averaged over two sites, at 8 cm depth (HFT3 plate, Campbell Scientific Inc., Logan, USA and HFP01 plate, Hukseflux, Delft, NLD), soil temperature (Ts), averaged over two sites, at 10 cm depth (TCAV Thermocouple probes, Campbell Scientific Inc., Logan, USA) and volumetric soil water content (SWC) at 10 cm (averaged over two sites) and 50 cm depth (CS616 water content reflectometer probes, Campbell Scientific Inc., Logan, USA).All instrumentation was powered by a remote area power system consisting of a diesel generator and a 24V battery bank inverter system
(Powermaker Ranger 4.5, Eniquest, QLD, AUS). An automated remote connection using a GSM modem (GPRS/ GSM Quadband Unimax Router and Ethernet modem, Maxon Australia Pty Ltd, Padstow, NSW, AUS) provided real time information on system status and ensured data acquisition on a daily basis. Additionally data were stored on an external CF (compact flash) cards which were interchanged on a monthly basis. A footprint analysis by Griebel et al. (2016) using the parameterisation of flux footprint predictions of Kljun et al. (2004) showed that the distribution of fluxes were relatively
homogeneous and that the whole footprint consisted of the same forest type and dominant tree species, and roughly uniform basal area. For further details see Griebel et al. (2016).

In February 2012, a custom-built profile system including an IRGA (Li-840, LI-COR, Lincoln, USA) and with six vertical layers (1, 2, 4, 8, 15 and 30 m) was installed to measure $CO_2$ concentrations of each layer in 2 min intervals (McHugh et al., 2016). A detailed description of the profile system can be found in McHugh et al. (2016). Due to technical problems with the
IRGA profile data were available from March to October 2012. Changes in the storage term between forest floor and EC-measurement point were calculated following McHugh et al. (2016) and Finnigan (2006). For periods of time when profile storage measurements were not available, ecosystem $CO_2$ fluxes were accounted for storage terms derived from single point calculations within the "OzFlux QC" data processing (Isaac et al., 2016). However, the contribution of storage term only marginally changed the magnitude of NEE (on average 2%) (Fig. S1 in supplementary materials).

 ## 2.3 Data processing

### 2.3.1 Quality Control

Quality assurance/ quality control (QA/QC) and eddy covariance flux corrections were performed on both available 10 Hz data and 30 minute covariance data. 10 Hz data was processed with Eddy Pro Version 6.2 (2016) including default statistical analysis (spike removal, drop-outs, absolute limits, skewness/kurtosis), low and high frequency correction (Massman, 2000;

 Moncrieff et al., 2005) and planar fit coordinate rotation (Wilczak et al., 2001). The calculated covariances from the 30 min averaging period were processed following the OzFlux standard protocol and open source code OzFluxQC version 2.9.6e (Isaac et al., 2016; OzFlux, 2016) using Anaconda Python version 2.7 (Continuum Analytics, Texas, USA). The procedure is described in detail in its own method paper by Isaac et al. (2016), as well as by Eamus et al. (2013) and Cleverly et al. (2013). In brief, the OzFluxQC procedure included quality control checks such as range checks in plausible limits, spike

 detection, dependency checks and manual rejection of date ranges of all measured variables (covariances and environmental variables) depending on site characteristics and based on visual revision of the data during the QA/QC procedure modified per month and year; linear corrections for calibration anomalies and sensor drift, 2D coordinate rotation (Lee et al., 2005), WPL correction (Webb et al., 1980), low and high frequency correction according to Massman (2001; 2000) and Moncrieff et al. (2005), conversion of virtual heat flux to sensible heat flux and correction of ground heat flux for heat storage in the

 soil layer above, addition of single point calculated or profile measurement derived storage term and calculation of fluxes from the quality controlled and corrected covariances. Extensive comparison between 10 Hz data processed with EddyPro and 30 min covariances processed with OzFluxQC showed that the planar fit correction versus 2D-coordinate rotation resulted in a 3% difference of fluxes. When this difference was accounted for in the OzFluxQC processed data set, both data sets were in very good agreement (slope: 1.01, intercept: 0.06, $R^2$: 0.90). Periods of data with low turbulence conditions,

 predominantly during night time, were excluded based on friction velocity (u*). Night time u* was filtered with yearly determined u*- thresholds using the change point detection method after Barr et al. (2013) and is described in detail in Isaac et al. (2016), Beringer et al. (2016b) and McHugh et al. (2016), this issue. Uncertainty in the u* threshold was estimated by generating a probability distribution for u* threshold and 95% confidence interval (CI) by bootstrapping the CPD method (1000 times randomly sampling of the data per year). Annual u* thresholds ranged from 0.53 to 0.66 m s$^{-1}$. Data gaps

occurred due to rainfall and occasional power failure and 60% of data were available over the three year period. Following QA/QC and night time u* filtering this was reduced to 37%, 49% and 49% in 2010, 2011 and 2012. From this quality filtered data were 64% day-time data and 26% night-time data.

### 2.3.2 Gap filling

Subsequent gap filling of data was done either with the Dynamic Integrated Gap filling and partitioning for OzFlux routine (DINGO v13) (Beringer et al., 2016b) or with the OzFluxQC procedure (Isaac et al., 2016), depending on the partitioning method selected (see below). However, both procedures have very similar data gap filling procedures and are described in detail in Beringer et al. (2016b) and Isaac et al. (2016). Small data gaps ($\leq$ 2 hrs) of continuous 30 min flux measurements and environmental variables were filled with linear interpolation.

In DINGO data gaps of environmental variables (air temperature, humidity, radiation, wind speed, atmospheric pressure and rainfall) > 2 hrs were gap filled: 1) from linear regressions with AWS (Automated Weather Stations) 30 min data records from the three nearest Bureau of Meteorology Australia (BoM) weather stations which were ranked after best correlation, 2) with spatially gridded meteorological daily satellite data at 0.1° resolution from the Australian Water Availability Project (AWAP, Raupach et al. (2009)) and in the unlikely event that gaps were still present after applying the above methods then a monthly diurnal means of measured climate variables were used. The frequency at which to perform the correlation analysis between flux tower data and AWS was set to use all available data. Soil temperature and soil moisture variables were gap filled using a simulation of the land surface using AWAP climate data and the CSIRO process-based land surface model BIOS2 at 0.05° resolution (see Haverd et al. (2013a)) adjusted to site observations. Following gap filling of environmental variables half-hourly NEE data were gap filled using a fast forward artificial neural network (FFNET ANN) with incoming shortwave solar radiation (Fsd), vapour pressure deficit (VPD), SWC, Ts, wind speed (Ws) and enhanced vegetation index (EVI) as input drivers according to Beringer et al. (2007); (2016b) and Papale and Valentini (2003). EVI was obtained from 16- day compositing periods of enhanced vegetation index (EVI) from MODIS (Moderate Resolution Imaging Spectroradiometer, see Huete et al. (2002)) as surrogate information of vegetation activity (i.e. leaf area index and growth) and interpolated to 30 min as proxy for production related to plant respiration. Frequency of gap filling using ANN was set to all available data.

In OzFluxQC data gaps of environmental variables > 2 hrs were gap filled: 1) with AWS as in DINGO, 2) using the regional Australian Community Climate Earth System 5 Simulator (ACCESS-R) numerical weather prediction (NWP) model at a resolution of 12.5 km run by the BoM (Isaac et al., 2016) and 3) ERA Interim (ERAI) data set from the European Centre for Medium Range Weather Forecasting (Dee et al., 2011) at 75 km resolution across Australia. Half-hourly NEE data were gap filled using the SOLO neural network (SOLO ANN) (Abramowitz, 2005; Hsu et al., 2002) with net radiation (Fn), ground heat flux (Fg), specific humidity (q), VPD, SWC, Ta and Ts as input drivers according to Isaac et al. (2016).

### 2.3.3 Partitioning and carbon flux definitions

The partitioning of NEE into its component fluxes GPP and ER was following the assumption of

$$NEE = ER - GPP \tag{1}$$

where day time NEE is the difference of GPP and ER, and night time NEE is equal to ER and hence, GPP being negligible/zero. We adopt the conventions in Chapin et al. (2006) where GPP and ER fluxes are designated with a positive sign. Negative NEE fluxes denote a net carbon flux from the atmosphere to the ecosystem, thus a net carbon uptake by the forest ecosystem.

One of the most common uncertainties in EC measurements can be an underestimation of night time NEE or ER as turbulent mixing is often lower or absent at night time which can lead to non-detectable vertical and horizontal advection of $CO_2$ within the canopy (Aubinet et al., 2012; Baldocchi, 2003; Goulden et al., 1996; van Gorsel et al., 2007). Although u* filtering is the most common correction for this underestimation error (Goulden et al., 1996), many studies have reported smaller estimates of ER from u* filtered and gap-filled EC-tower data compared to those from chamber measurements of soil, leaf and stem respiration (Keith et al., 2009; Lavigne et al., 1997; Law et al., 1999; Phillips et al., 2010; Speckman et al., 2015). Although no independent up-scaled ER estimates from chamber measurements were available from our study site, we used independent daily soil respiration data from a separate study at the same study site (Hinko-Najera, 2016, unpublished)to visually compare its relative contribution to daily tower ER estimates derived from four different data selection and subsequent partitioning methods to reduce a potential underestimation of ER (see supplementary material Fig. S1, S2 and S3). We ran an ensemble of different partitioning methods (using either DINGO or OzFluxQC routines) and u* based night time filters on NEE fluxes including the storage term only to evaluate variation of ecosystem carbon fluxes

depending on partitioning and filter method used. An overview of partitioning methods, estimated annual sums and their variation is given in supplementary materials in Table S1,S2 and Fig. S4, and briefly explained here: We used three different partitioning methods to estimate gross ecosystem carbon fluxes: 1) night time approach after the Lloyd and Taylor temperature response function (Lloyd and Taylor, 1994; Reichstein et al., 2005) with Ta as input driver using a window size of 15 days with an overlapping of 10 days in OzFluxQC, 2) night time approach using ANN: (2a) FFNET NN with Ts, Ta,

SWC and EVI as input drivers and a window size of all available data in DINGO (Beringer et al., 2016b) or (2b) SOLO NN with Ta, Ts and SWC as input drives and a window size of one year in OzFluxQC (2b) (Isaac et al., 2016) and 3) day time approach using the light response function according to (Lasslop et al., 2010) using either (3a) DINGO or (3b) OzFluxQC with a window size of 15 days with an overlapping of 10 days. Detailed descriptions of functions and routines used within the DINGO and OzFluxQC routines are given in Beringer et al. (2016b) and Isaac et al. (2016). For the methods using the

night time approach the u* filter after the u* threshold was applied to non gap filled (quality controlled observations only) night time (Fsd $<10$ W m$^{-2}$) NEE flux data. Differences between DINGO (2a) and OzFluxQC (1, 2b) partitioning methods here is that DINGO was set to use u* filtered night time data from the first three hours after sunset only (Fsd $<10$ W m$^{-2}$) while with OzFluxQC three night time selections have been applied to u* filtered night time data: all u* filtered night time NEE data, first three hours after sunset of u* filtered NEE data, and a variable daily window size using all night-time data

above the u* threshold from sunset onwards until u* falls below the u* threshold (Eva van Gorsel, personal communication). The selection of the first three hours after sunset is based on an extensive study in a wet temperate eucalypt forest from van Gorsel et al. (2008); (2007) who demonstrated that ER was at maximum in the early evening hours when the canopy is still coupled with the atmosphere. For the ANN methods (2a) and (2b) estimated night time ER was extrapolated to day time ER. The final NEE flux was then constructed from gap filled day time data (Fsd $>= 10$ W m$^{-2}$) and estimated ER at night time.

GPP was then subsequently estimated with Eq. (1). For the methods using the daytime approach a light response curve was fitted to day time NEE to estimate GPP and subsequently ER across day and night time.

## 2.4 Uncertainty analysis and analysis of environmental drivers

We performed an uncertainty analysis as described in McHugh et al. (2016) which includes an uncertainty estimation of combined random and model error (Hollinger and Richardson, 2005; Keith et al., 2009) (supplementary material Table S3)

and the effect of uncertainties in u* thresholds on annual NEE estimates by using the lower (5%) and upper (95%) confidence interval of the probability distribution of the mean u* threshold (Barr et al., 2013) (supplementary material Table S4). The uncertainty introduced due to random (measurement) and model error was small for 2010 and 2012 (4% and 6% of annual NEE estimate) and slightly higher for 2011 (10% of annual NEE estimate) (Table S3). Estimation of the uncertainties in u* thresholds on annual NEE were small for the lower u* uncertainty bound (5% CI) ranging from -19 g C m$^{-2}$ yr$^{-1}$ in 2012 to 26 g C m$^{-2}$ yr$^{-1}$ in 2011 (1-2% of annual NEE estimate). We could not estimate an upper u* uncertainty bound (95% CI) as data availability was insufficient (Table S4).

To analyse environmental controls (Fsd, Ta, VPD and SWC) on the seasonal and inter-annual variability of day time NEE a rectangular hyperbolic light response curve (LRC) or *Michaelis-Menten* equation (Carrara et al., 2004; Falge et al., 2001; Flanagan et al., 2002; Lasslop et al., 2010; Michaelis and Menten, 1913) was used to determine the dependency of daily means of quality controlled half-hourly non gap filled midday NEE (hours 11:00 – 13:00) on Fsd:

$$NEE_{midday} = \frac{\alpha \times Fsd \times \beta}{\alpha \times Fsd + \beta} \tag{2}$$

where Fsd is the incoming solar radiation (W m$^{-2}$), $\alpha$ is the slope of LRC or the canopy light utilization efficiency (μmol $CO_2$ J$^{-1}$) and $\beta$ the maximum NEE or  uptake rate of the canopy at light saturation (μmol $CO_2$ m$^{-2}$ s$^{-1}$). Residuals of the LRC were then used to analyse the dependency of midday NEE on either Ta or VPD (given the strong correlation between VPD on Ta) (Carrara et al., 2004; Chen et al., 2002) with linear and non linear regressions, i.e. exponential temperature sensitivity functions according to Lloyd and Taylor (1994) for Ta (see equation 3 below) and a logarithmic power model according to Chen et al. (2002) for VPD. However, for both, Ta and VPD, linear relationships resulted in the best fits whereas non linear regressions consistently resulted in arbitrary or insignificant parameter estimates. The influence of SWC on day time NEE was tested with linear regressions between SWC and residuals of LRC and 2$^{nd}$ residuals from the linear relationships between LRC residuals and Ta as temperature and soil moisture are often negatively correlated in this forest ecosystem (Hinko-Najera et al., 2015). To analyse the effect of Ta, VPD and SWC on LRC data was divided into Ta -, VPD - and SWC bins.

For the analysis of analyse environmental controls (Ta, SWC) on the seasonal and inter-annual variability of night time NEE an Arrhenius-type model function (LT) according to Lloyd and Taylor (1994) was applied to determine the temperature sensitivity of daily means of half-hourly quality controlled non gap filled and u* filtered night time NEE:

$$NEE_{NT} = R_{\text{ref}} \times exp\left(E \times \left(1/(T_{\text{ref}} - T_0) - 1/(Ta - T_0)\right)\right) \qquad (3)$$

where $R_{\text{ref}}$ the basal respiration rate ($\mu$mol $CO_2$ m$^{-2}$ s$^{-1}$) at the reference soil temperature of 10°C ($T_{\text{ref}}$ = 283.15 K), E the activation energy related parameter, $T_0$ a fixed temperature parameter ($T_0$ = 227.13 K) according to Lloyd and Taylor (1994) and Ta is the air temperature (K). Residuals of LT were then used to analyse the dependency of night time NEE on SWC with linear regressions. To analyse the effect of SWC on LT data was divided into SWC bins.

All subsequent data manipulation and statistical analyses were done using R version 3.3.2 (R Core Team, 2016) and for gap filled times series we chose the night time partitioning approach from the DINGO output. Differences in seasonal and inter-annual variations of daily means were tested with Kruskal-Wallis rank sum test and Dunn's Test.

### 3 Results

#### 3.1 Seasonal and inter-annual variation in environmental variables

Seasonal and inter-annual pattern in rainfall varied markedly across the three year period (Fig. 1b, 2f). In the first two years unusually high rainfall was observed during the occurrence of two strong La Niña events. Annual rainfall in 2010 and 2011 was 43% and 22% above the long-term mean annual rainfall from the nearest BOM station (Daylesford) which is representative for the study site as an adjustment was less than 1% (Table 1, Table 2). Most of the anomalous rainfall occurred between August 2010 and February 2011 with a 2-fold increase in rainfall during spring 2010 (S-O-N) and a 3-fold increase in rainfall in summer 2010/11 (D-J-F, Fig. 1b). While the annual rainfall in 2012 was close to the long-term mean annual rainfall (Table 1, Table 2), monthly rainfall showed a distinct pattern with the above long-term mean rainfall in February 2012 (2-fold increase) and winter 2012 (J-J-A, +30%) but below long-term mean rainfall from spring 2012 (S-O-N, -37%) onwards (Fig 2.1b, 2.2f).

SWC at 10 cm soil depth generally varied strongly with seasons and was highest during winter with a daily maximum of 0.36 cm$^3$ cm$^{-3}$ observed in August 2012 and decreased towards summer reaching a daily minimum of 0.12 cm$^3$ cm$^{-3}$ in

February 2012 (Fig. 2e). Seasonal variability of SWC was more pronounced in 2012 (CV = 25%) than in 2010 (CV = 18%) as high rainfall led to an absence of a dry period during summer 2010/11 (D-J-F) and SWC remained relatively stable and high throughout 2011 (CV = 13%)..

Fsd, Ta, Ts and VPD showed a strong seasonality with maximum values during summer months and minimum values during winter months (Fig. 2b, c, d). Mean daily Fsd was 278 W m$^{-2}$ in summer (maximum of 411 W m$^{-2}$ in January 2011) and 81 W m$^{-2}$ in winter (minimum of 5 W m$^{-2}$ in May 2011, Fig 2.2b). Daily mean temperatures ranged from 2.0 ºC in winter (J-J-A) to 28.2 ºC in summer (D-J-F) for Ta and from 4.8 ºC and 23.6 ºC for Ts (Fig. 2c). Daily maximum midday VPD of 3.41 kPa was observed in January 2012 (Fig. 2d). Inter-annual variation of Fsd, Ta and Ts were marginal (Fig. 2 b,c) while mean

annual VPD was significantly (p <0.001) higher in 2012 (0.47 kPa) compared to 2010 and 2011 (0.36 kPa and 0.32 kPa, Table 2).

### 3.2 Seasonal and inter-annual variation of $CO_2$ ecosystem fluxes

A pronounced seasonal pattern was observed in ER (CV = 39%), but less so in GPP (CV = 27%) and NEE (CV = 24%, Fig. 2a, 3, 4).

In general, the forest showed near-continuous net carbon uptake (negative NEE) throughout the three year period, with highest net uptake rates during summer and spring months (seasonal daily means of NEE: -4.07 and -3.98 g C m$^{-2}$ d$^{-1}$) and lowest rates during winter and autumn months (seasonal daily mean NEE: -2.86 and -2.62 g C m$^{-2}$ d$^{-1}$, Fig. 2a, 3). Only in December 2010 and January 2011 NEE became distinctly positive (net carbon loss) for a short period (3 to 4 consecutive days) due to an increase in ER in December 2010, and a decrease in GPP in January 2011 (by ~40%) in response to

considerably high rainfall and limited solar input. Seasonal variability in NEE was noticeably less pronounced in 2011 (CV = 9 %) than in 2010 (p <0.001, CV = 38 %) and 2012 (p <0.001, CV = 21 %, Fig. 4a). Inter-annual variation of NEE was high (CV = 15 %) and significant (p <0.001) with most evident differences during winter and autumn (Table 2, Fig. 4a).

ER was greatest during summer (seasonal daily mean of 5.77 g C m$^{-2}$ d$^{-1}$), of similar magnitude in autumn and spring (4.34 g C m$^{-2}$ d$^{-1}$ and 4.23 g C m$^{-2}$ d$^{-1}$, respectively) and lowest during winter (2.05 g C m$^{-2}$ d$^{-1}$). Similarly to NEE, GPP was greatest

during summer (seasonal daily mean of 9.85 g C m$^{-2}$ d$^{-1}$), followed by spring (8.20 g C m$^{-2}$ d$^{-1}$), autumn (6.96 g C m$^{-2}$ d$^{-1}$), and least during winter (4.91g C m$^{-2}$ d$^{-1}$).

Overall inter-annual variation for ER was moderate (CV = 9 %) and significant for all seasons except spring. ER was significantly lower in summer 2012 compared to summer months in 2010 and 2011, as was winter 2012 compared to winters 2010 and 2011 (Fig. 4b). Overall, GPP estimates did not vary between years (CV < 3 %) and only in winter 2010 GPP significantly differed from the winters of 2011 and 2012 (Fig. 4c). The ER/GPP ratio was highest during autumn and summer (0.62 and 0.59) and lowest during spring and winter (0.52 and 0.42).

### 3.3 Environmental drivers of $CO_2$ ecosystem fluxes

Results of the LRC fits and linear fits with T and VPD per year and seasons are presented in Table 3. Overall, Fsd explained 25% of the temporal variability in midday NEE (Fig. 5a) which did not considerable vary between observation years. Similarly both T and VPD explained about 18% or 23% of the overall temporal variability in midday NEE (Fig. 5b,c) and again no considerable inter-annual differences were determined. However, a clear distinct seasonal pattern was shown in the dependencies of midday NEE on Fsd, T and VPD when coefficients of determinations are plotted for each month (Fig. 6a). While Fsd was the dominant environmental driver during mid/ late autumn and winter months (36-47%, mean = 42%), Ta and VPD were the main controlling environmental variables during spring, summer and early autumn months (23-56%, mean = 40% for Ta and 15-48%, mean = 31% for VPD). LRCs fitted for various Ta bins and VPD bins (Table 4) show a strong decrease in the net carbon uptake at temperatures above 20°C or VPD values above 1.2 kPa. A clear differentiation between Ta and VPD was very difficult because of the strong correlation between VPD and Ta. While overall and inter-annual variability in residuals of midday NEE were marginally better explained by VPD than Ta, variability of midday NEE residuals from spring to early autumn correlated stronger with changes in Ta than VPD. No apparent influence of SWC on residuals of NEE or LRCs were determined (Table 3c, 4).

Results of LT and linear fits with SWC per year and seasons are presented in Table 5. Overall 36% of the temporal variability in u* filtered night time NEE was explained by Ta (Fig. 5d) which varied from 30% in 2012 to 31% in 2010 and 49% in 2011. On seasonal time scales the dependence of night time NEE on Ta strongly varied being greatest during spring (39-44%, mean = 42%) followed by summer months (31-45%, mean = 38%) and lowest during autumn months (15-30%, mean = 24%) (Fig. 6b). No significant relationships could be determined during winter months where greater data gaps occurred compared to other months. Neither LT fitted for SWC bins (Table 5) nor linear relationships between SWC and LT

residuals showed an influence of SWC on night time NEE or its temperature sensitivity with the exception of the winter month July where night time NEE decreased with increasing SWC ($R^2$ = 0.26 ***).

### 3.4 Annual estimates of NEE, GPP and ER and associated uncertainties

From the various partitioning methods applied the DINGO output with the night time data approach using NN and u* filtered early evenings hours resulted in ER estimates most consistent with the soil respiration data – in terms of the relative contribution of soil respiration exceeding 1 was minimal (see supplementary material Fig. S2a). However, the day time approach partitioning method using the light response curve with the DINGO routine (3a) (Fig. S3a) yielded similar results followed by the night time approach using the Lloyd and Taylor temperature response function (1) when night time fluxes

only were filtered after u* threshold (Fig. S1c). Overall, coefficient of variation of annual estimates of ER derived from different partitioning methods ranged from 8 to 10% while variation of annual estimates of NEE and GPP were small (4-7% and 2-4%) (see supplementary materials Fig. S4, Table S2).

The forest was a considerable and continuous carbon sink during the three year study period with a mean NEE of -1234 ± 109 g C m$^{-2}$ yr$^{-1}$. Estimates of annual NEE increased from -1046 g C m$^{-2}$ yr$^{-1}$ in 2010 to -1424 g C m$^{-2}$ yr$^{-1}$ in 2012 (Table 2).

Estimates for annual ER were similar for 2010 and 2011, whereas in 2012 annual ER decreased (Table 2). Annual GPP estimates were slightly higher in 2010 compared to other years and similar for 2011 and 2012 (Table 2). ER was on average 55% of GPP, but this ranged between 61% (2010) and 49% (2012) (Table 2).

### 4 Discussion

### 4.1 Seasonal variability of $CO_2$ ecosystem fluxes

Gross and net $CO_2$ ecosystem fluxes showed strong seasonality in this dry temperate eucalypt forest. On a seasonal time scale, GPP exceeded ER almost continually, even in winter, thus NEE showed a net carbon uptake across all seasons. Daily minimum and maximum rates of ER, and daily minimum rates of GPP were within the reported range for temperate evergreen coniferous and deciduous broadleaved forests compiled by Falge et al. (2002). Although daily maximum GPP rate at our forest site (14.7 g C m$^{-2}$ d$^{-1}$) were comparable with those from temperate evergreen coniferous forests (16.6-26.3 g C

m$^{-2}$ d$^{-1}$), they were much lower than those reported for temperate deciduous broadleaved forests (22.4-31.0 g C m$^{-2}$ d$^{-1}$) during growing seasons (Falge et al., 2002).

Both GPP and ER peaked during summer and were lowest in winter which is similarly typical for temperate evergreen *coniferous* forests (Baldocchi, 2008; Baldocchi and Valentini, 2004). However, during spring the increase in ER lagged behind that of GPP, an occurrence similarly typical for temperate *deciduous* broadleaved forests (Baldocchi, 2008;

Baldocchi and Valentini, 2004). Thus, NEE peaked in early spring and again in summer. A likely explanation for the delayed increase in ER as compared to GPP in springtime could be a limitation of microbial decomposition due to lower temperatures or substrate supply. Partitioning of soil respiration into its component fluxes of heterotrophic (microbial) and belowground autotrophic (plant) respiration in an earlier study (Hinko-Najera et al., 2015) showed that heterotrophic respiration was low during springtime but increased and peaked during (late) summer months corresponding to when total

soil respiration fluxes were greatest (Fig. S2a). Similar seasonal variability in GPP and ER was observed in a wet temperate eucalypt forest, however, NEE peaked only during spring in this forest as GPP was limited by water availability during dry summer period (Kilinc et al., 2013). These phenomena highlight that carbon exchange dynamics in this dry temperate eucalypt forest are different in their seasonal behaviour from temperate deciduous broadleaved or evergreen coniferous forests in the Northern Hemisphere (Baldocchi, 2008; Baldocchi and Valentini, 2004; Falge et al., 2002), so the latter should

not be used as analogues.

### 4.2 Environmental drivers of CO$_2$ ecosystem fluxes

As indicated above the overall main environmental drivers of NEE were radiation and temperature. Temporal variability of day time NEE was equally strong influenced by radiation and temperature and/or VPD which did not vary throughout the three year study period. Interestingly, a strong seasonal variability of environmental controls on NEE was revealed. While

radiation was the dominant environmental driver of day time NEE in mid to late autumn and winter, temperature was the main environmental control from spring to early autumn. In accordance with day time NEE, temperature sensitivity of night time NEE followed a distinct seasonal pattern with greater influence of Ta on night time NEE during spring and summer months. An indication of a limitation of respiration due to high water content in July is, however, precautious as data availability was lowest (<50%) during this month.

Dry temperate eucalypt forests, like most of Australia's forests, are generally characterised by dry summer periods and thus are greatly influenced by changes in water availability (Haverd et al., 2013a). However, environmental drivers related to water availability such as VPD and SWC had either a small or no influence on day time NEE fluxes during summer months and night time NEE fluxes. Thus, there was no apparent water limitation on carbon dynamics during our study period in this forest. This is in contrast with findings from Keith et al. (2012), van Gorsel et al. (2013) and Kilinc et al. (2013) for wet

temperate eucalypt forests. In all these studies ecosystem carbon fluxes were limited by VPD and/or SWC during dry summer months with greatest effects on day time NEE than night time NEE. However this can be explained by the anomalously high rainfall at our forest site due to strong La Niña events from mid 2010 to early 2011 and early 2012 (BoM, 2012) during most of the study period. The lack of SWC influence on carbon fluxes until the end of 2012 was also evident in concurrent studies on soil respiration dynamics at the same study site where SWC did not decline below a certain threshold

to be limiting soil respiration (Hinko-Najera et al., 2015).

### 4.3 Annual carbon balance and uncertainties

The dry temperate eucalypt forest was a very strong and continuous carbon sink for all three years. Our mean annual ER/GPP ratio (~0.55) was lower than the mean ER/GPP ratio of 0.76 for Australian ecosystems (Beringer et al., 2016a), the 0.80 reported for temperate forests (Janssens et al., 2001; Luyssaert et al., 2007) and the 0.77 derived from a global data base

(Baldocchi, 2008). Moreover, our annual estimates for NEE are greater than published estimates for forest ecosystems around the globe collated by Baldocchi et al. (2001) or Luyssaert et al. (2007), and at the upper end of the probability distribution for sites within the global FLUXNET network (Baldocchi, 2014).

That temperate eucalypt forests in Australia exhibit large carbon uptake has been shown in wet temperate eucalypt forests (>1000 mm annual rainfall) only recently.A tall old growth *E. regnans* forest had an NEE of -930 g C m$^{-2}$ yr$^{-1}$ (Beringer et

al., 2016a). A wet temperate *E. delegatensis* forest near Tumbarumba (Keith et al., 2009; 2012; van Gorsel et al., 2013) was a strong sink of ~ 900 g C m$^{-2}$ yr$^{-1}$ during years with average annual rainfall (~ 1400 mm), but this sink was reduced (~ 750 g C m$^{-2}$ yr$^{-1}$) during the above average rainfall (~ 2000-2200 mm) years of 2010 and 2011 (van Gorsel et al., 2013). Overcast conditions and thus reduced incoming solar radiation explained this reduced sink (van Gorsel et al., 2013). However, NEE estimates we report for our study site are higher than those reported from the Tumbarumba eucalypt forest during the same

years of above average rainfall. A possible explanation for the greater net carbon uptake estimates in our dry temperate eucalypt forest might be the higher leaf area index (~ 1.8) (Griebel et al., 2016; Moore, 2011) than in the wet temperate eucalypt forest (~ 1.4) near Tumbarumba, conferring a higher canopy photosynthetic capacity. Our study supports the conclusion of Keith et al. (2009) that temperate eucalypt forests have a high carbon uptake potential because they are evergreen and as such photosynthetic carbon uptake can occur throughout the year when conditions are favourable. Indeed, a

detailed study of stem and canopy growth dynamics at our study site demonstrated that the trees are in fact growing all year, with canopy expansion observed mainly in summer and early autumn, and the stem growth mainly in spring and autumn, but also to a lesser degree in winter (Griebel et al., 2017).Another possible explanation for the higher net carbon uptake estimates at our study site is the absence of summer dry periods and a stimulation of growth due to the high rainfall. Prior to the period of high rainfall in 2010-2012 forests throughout temperate Australia experienced a decade long drought that

negatively affected NEE and NPP (Haverd et al., 2013b). Keith et al. (2012) and Kilinc et al. (2013) reported as well that drought conditions in the wet temperate eucalypt forest strongly reduced NEE by having a greater negative effect on GPP than ER. Therefore it is likely that the onset of high rainfall in winter/early spring 2010 led to favourable conditions for growth and high carbon uptake given the opportunistic behaviour of eucalypts to changing environmental conditions (Jacobs, 1955; Keith, 1997).

However, the possibility of a remaining underestimation of ER cannot be excluded although accounted for during data processing and partitioning and by the addition of storage term. We observed low night time fluxes of NEE, hence ER, which indicated a decoupling within the forest canopy and thus advection fluxes during night. Hence it is possible that the high NEE reported here are partly due to an underestimation of ER due to advection. However, such an error is likely to be a systematic one, meaning that an overestimation of NEE would have consistently occurred during the different measurement

years and seasons. Results from a recent study to validate more recent annual NEE estimates with a biomass inventory (biomass increment and carbon content) indicated that the NEE estimates from the EC measurements were systematically 30% greater than the NPP of the tree biomass from inventory methods (Bennett, 2016, unpublished). It is a reasonable assumption that tree NPP contributed the majority to NEE and as such the inventory data would be a confirmation that the flux tower is underestimating ER, resulting in a greater NEE. However, the underestimation was indeed of a similar

magnitude in each of the measurement years and confirmed the high carbon uptake rate of this dry temperate eucalypt forest when compared to other ecosystems.

Our data indicate that the main environmental controls for day time NEE (radiation and temperature) and night time NEE (temperature) did not vary between years. We observed moderate variations in NEE amongst the three years, with an increase in NEE from 2010 through to 2012. The high rainfall in late 2010 and early 2011 most likely led to favourable

forest growth conditions throughout 2011 and a stronger increase in GPP rather than ER and thus an increase in NEE from 2010 to 2011, most evident in autumn and winter. This 2011 increase in NEE is in accordance with the observed 2011 global sink anomaly (Haverd et al., 2013a; 2016; Poulter et al., 2014) which has been mainly attributed to semi-arid ecosystems in Australia (Eamus and Cleverly, 2015; Haverd et al., 2013a). Hence, our results indicate that the global sink anomaly was not only limited to semi-arid ecosystems. The further increase in NEE from 2011 to 2012 was indicated by a reduction in ER as

GPP remained steady. Rainfall was lower in 2012 as compared to the previous two years and hence, soil water content decreased towards the end of 2012 (summer) likely influencing ER but not GPP. Given that ER is often dominated by soil respiration, this pattern is in agreement with findings on soil respiration patterns from concurrent studies in the same forest (Hinko-Najera et al., 2015) where low soil water contents led to a decrease in soil respiration.

**5 Conclusion**

Temperate eucalypt forests are underrepresented in global assessments concerning terrestrial/ forest carbon dynamics and productivity (Baldocchi, 2008; Falge et al., 2002; Luyssaert et al., 2007) and so far no data has been available on ecosystem carbon exchange dynamics from dry temperate eucalypt forests. This study shows that not only wet temperate eucalypt forests but also dry temperate eucalypt forests have a large carbon uptake potential, particular during above average rainfall, and thus adds further evidence that temperate eucalypt forests are strong carbon sinks during favourable conditions (Keith et

al., 2009). Furthermore, carbon dynamics in this dry temperate eucalypt forest, similar to other temperate eucalypt forests, do differ in their seasonal behaviour compared to temperate evergreen coniferous and deciduous broadleaved counterparts forest in the Northern Hemisphere. The evergreen nature of the trees, coupled with mild winter temperatures allow for all year round physiological activity of eucalypts, which can lead to a continuous growth throughout the year. When this is coupled

with high rainfall in the warmer summer months it can lead to very large carbon uptake. However, long term measurements

over multiple years are required to evaluate the net carbon sink strength of dry temperate eucalypt forests further particularly in years with drought conditions, a scenario predicted to increase in occurrence and intensify in south-eastern Australia (Christensen et al., 2013; CSIRO, 2012). Furthermore studies using alternative approaches, for example independent up-scaled ER estimates from component flux measurements (Keith et al., 2009; Lavigne et al., 1997; Law et al., 1999; Phillips et al., 2010; Speckman et al., 2015) are needed to account for underestimation in ER due to advection fluxes and to validate

measurements from EC flux tower.

**Author contribution.** N. Hinko-Najera, S.K. Arndt, S.J. Livesley and J. Beringer designed the experiment. Field work was primarily carried out by N. Hinko-Najera with help from I. McHugh and J. Beringer. Data preparation and analysis was primarily performed by N. Hinko-Najera with contribution from P. Isaac, C.Ewenz and I. McHugh (quality control), J.

Beringer (DINGO), C.Ewenz and E. van Gorsel (partitioning), and J-F Exbrayat (analysis of environmental drivers). N. Hinko-Najera prepared the manuscript with contributions from all co-authors.

**Acknowledgements.** The study was supported by funding from the Terrestrial Ecosystem Research Network (TERN) Australian Supersite Network, the TERN OzFlux Network, the Australian Research Council (ARC) grants LE0882936 and

DP120101735 and the Victorian Department of Environment Land, Water and Planning Integrated Forest Ecosystem Research program. We would like to thank Julio Najera and Darren Hocking for site and instrument maintenance. J. Beringer is funded under an ARC Future Fellowship (FT110100602).

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

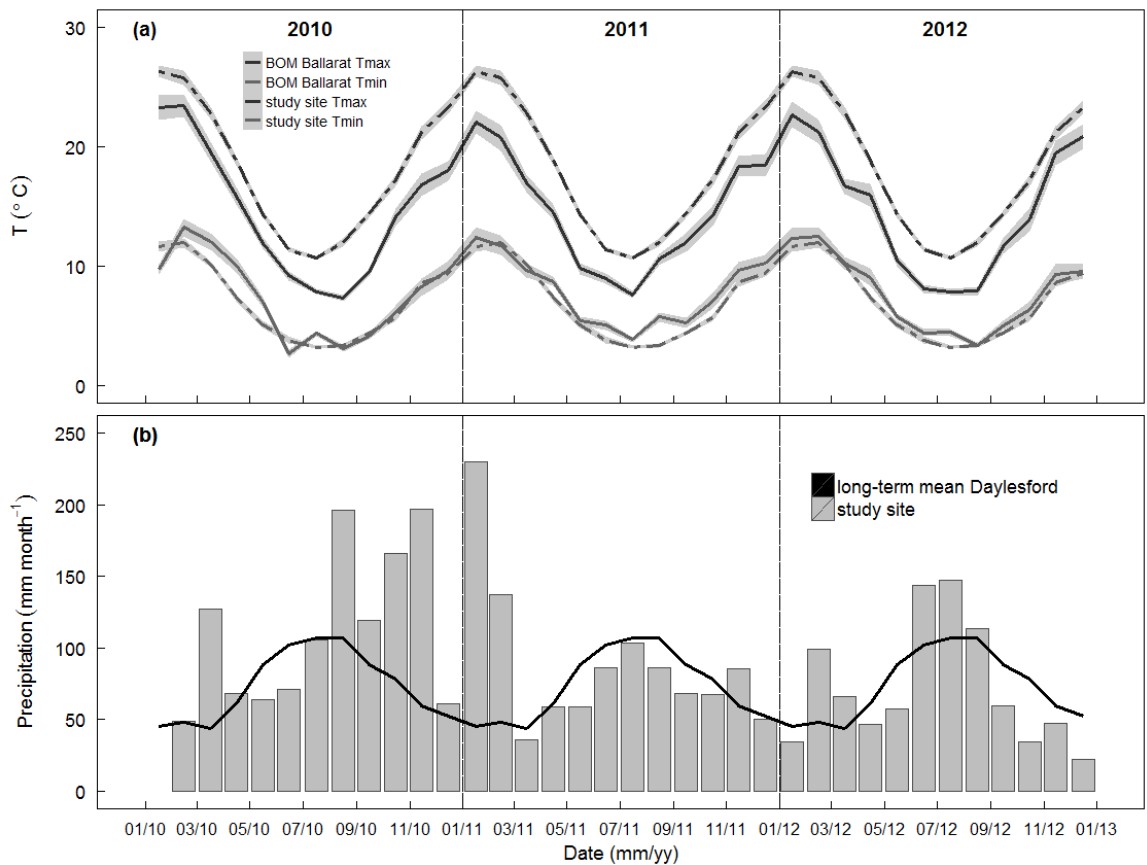

**Figure 1: Climate time series of (a) monthly averages of minimum (grey lines) and maximum (black lines) air temperatures from the study site from 2010 to 2012 (solid lines) and from the BoM station Ballarat from 2001 to 2013 (dashed lines), shaded areas indicate ±1SE, and (b) monthly rainfall at Wombat State Forest OzFlux EC tower site from 2010 to 2012 (grey bars) and 114 year (1901-2014) long-term monthly mean rainfall at BoM station Daylesford (black line)**


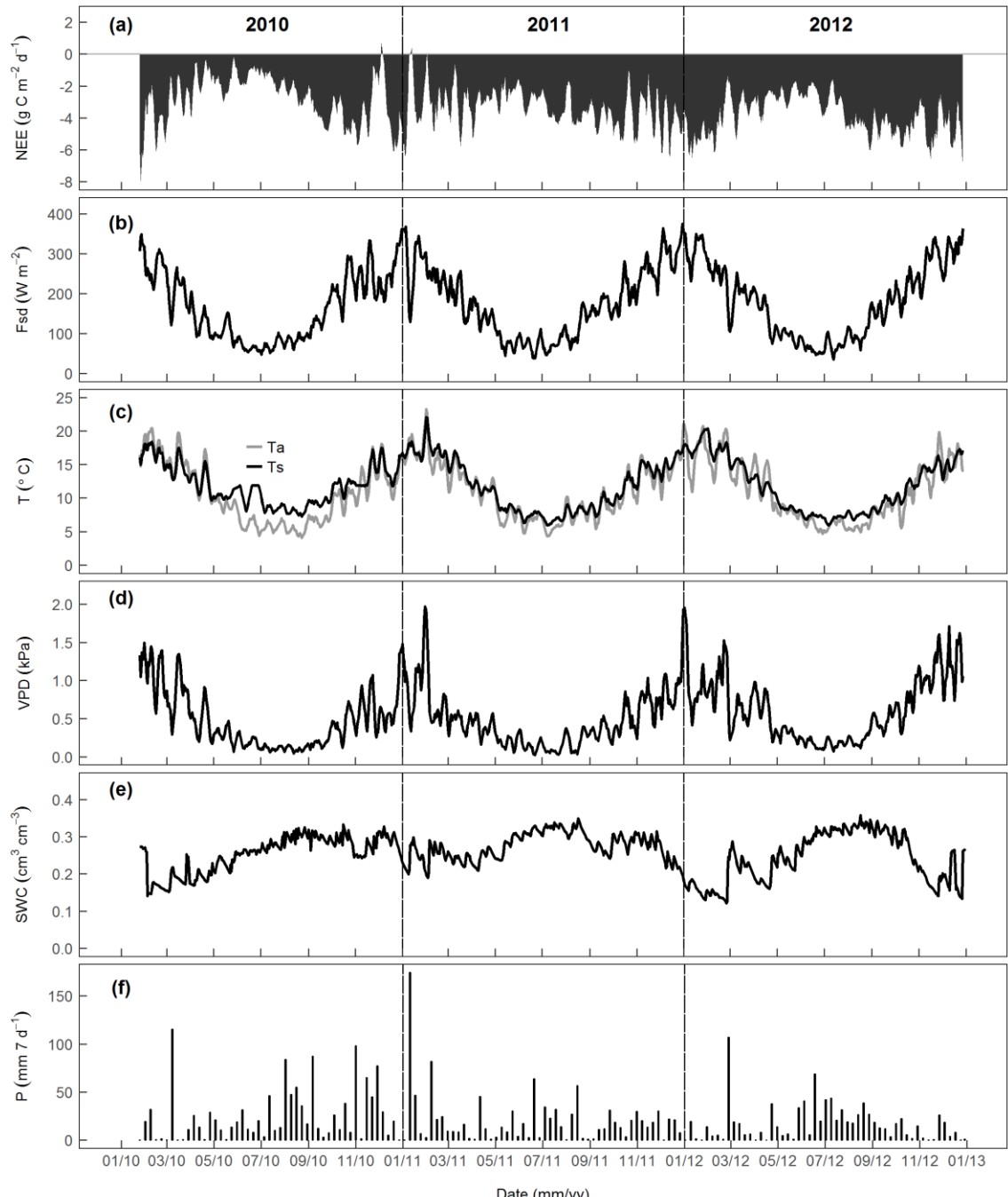


**Figure 2: Time series of (a) 7-day running means of daily total net ecosystem exchange (NEE), 7-day running means of daily averages of (b) incoming solar radiation (Fsd), (c) air (Ta) and soil (Ts) temperature, and (d) 7-day running means of mean midday (11:00-13:00) vapour pressure deficit (VPD), (e) daily averages of volumetric soil water content (SWC) and (f) 7-day sums of rainfall (P) from 2010 to 2012**


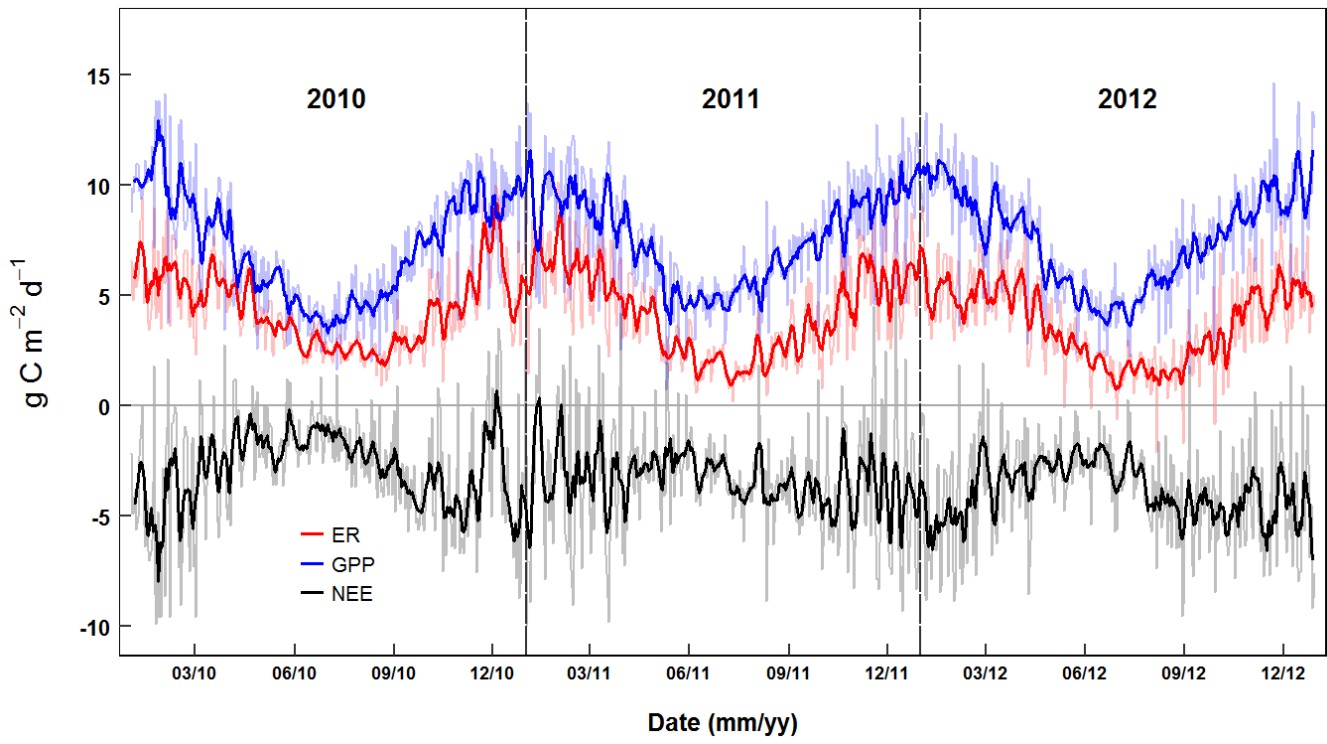

**Figure 3: Ecosystem carbon fluxes of the Wombat State forest OzFlux site from 2010 to 2012: ecosystem respiration (ER, red lines), gross primary productivity (GPP, blue lines) and net ecosystem carbon exchange (NEE, black lines), displayed is the output of DINGO partitioning method (2a) using the night time data approach with NN and early evening hours selection with daily totals (g C m$^{-2}$ d$^{-1}$) of ecosystem carbon fluxes (shaded lines) and 7-day running means of daily totals (bold lines) for better illustration.**


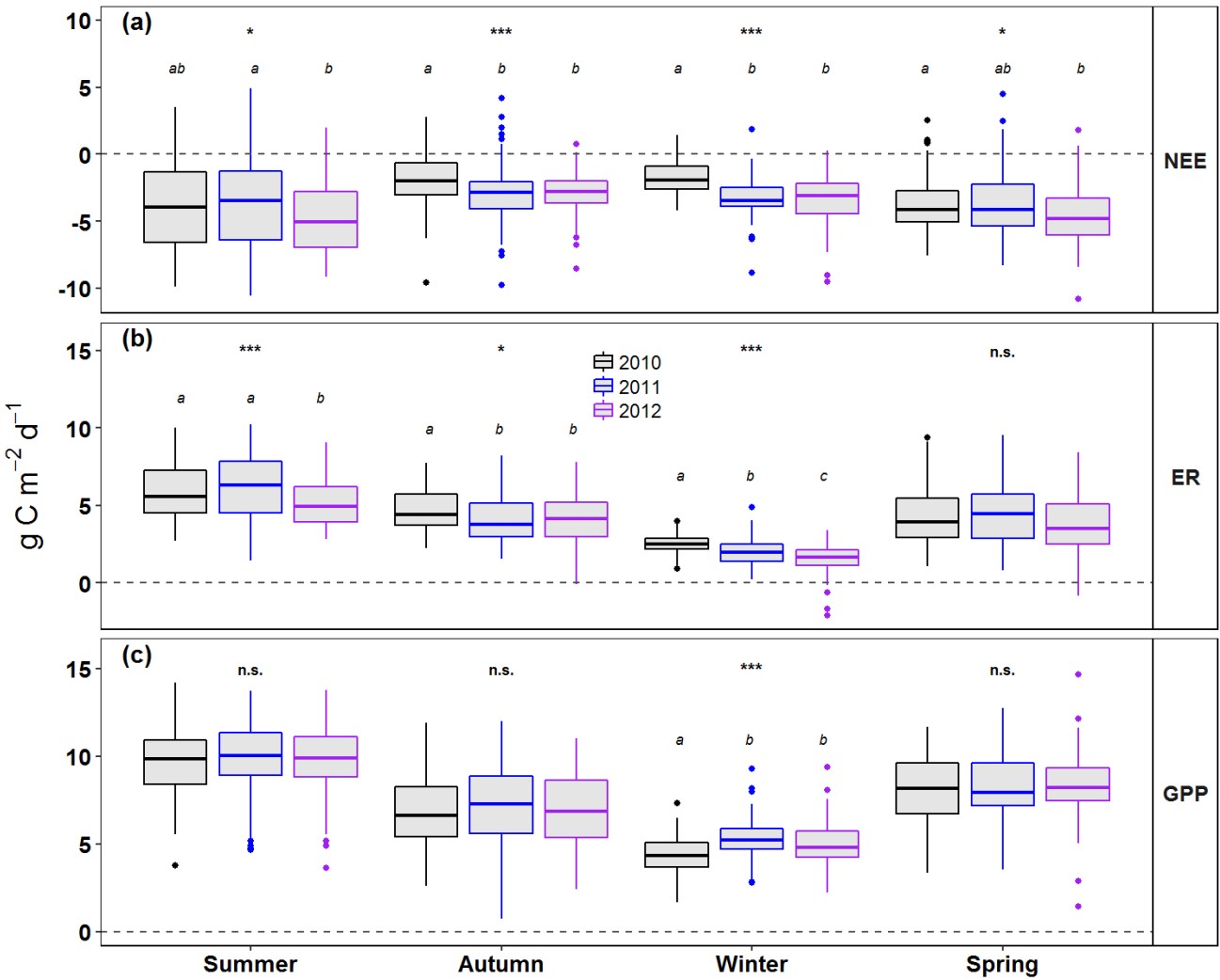


**Figure 4: Box- and whisker plots of daily averages of a) NEE, b) ER and c) GPP for years and seasons; inter-annual differences are displayed for each seasons with p-values (significance level p <0.05), letters indicate year to year differences**

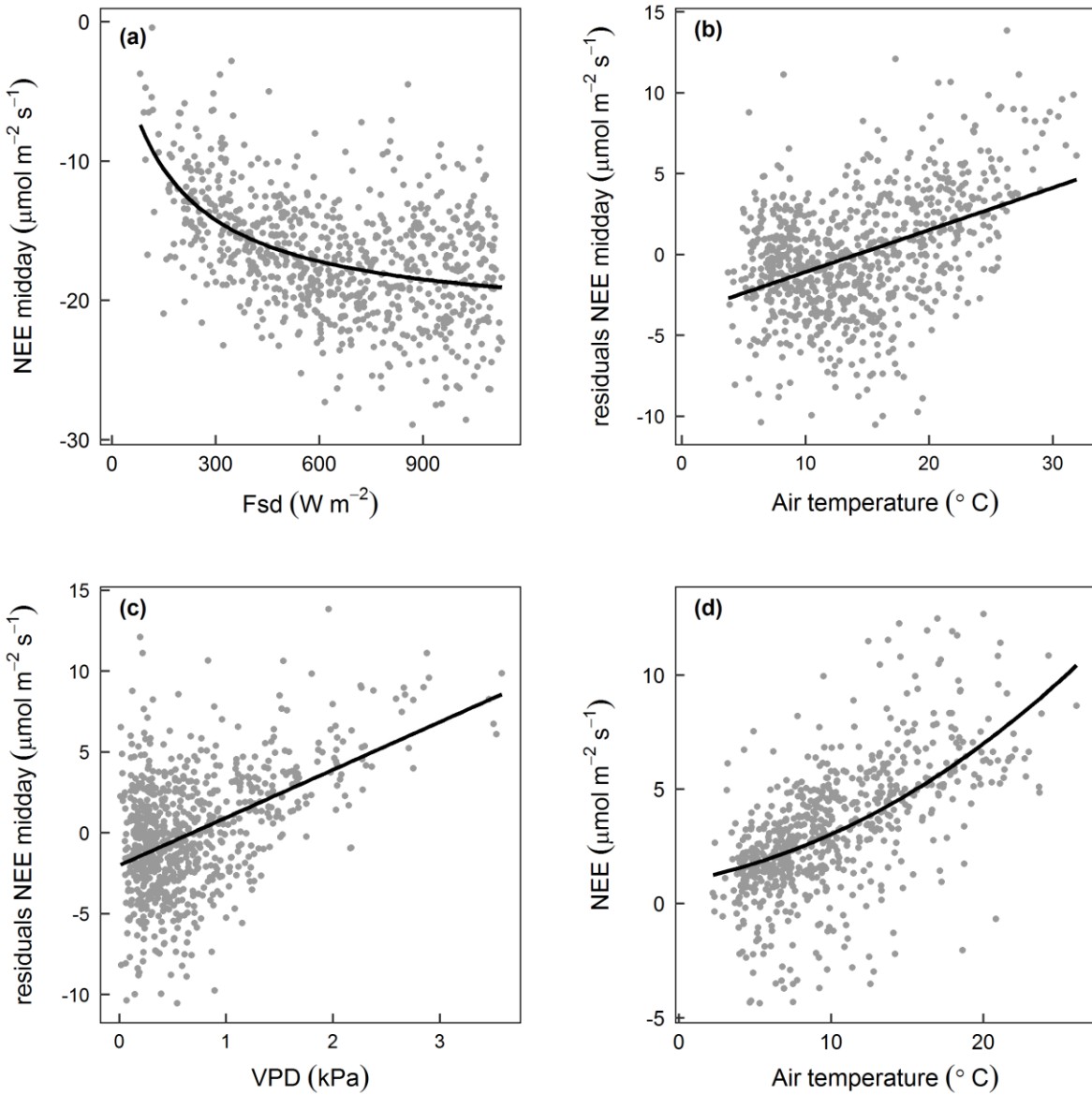

**Figure 5:** (a-c) Relationship between daily means of midday NEE and (a) incoming solar radiation (Fsd) as a rectangular hyperbolic light response curve (LRC), linear fits between residuals of LRC and (b) air temperature (Ta) or (c) vapour pressure deficit (VPD) and (d) relationship between daily means of u* filtered night time NEE and air temperature as temperature (Ta) response function after Lloyd and Taylor (1994), $R^2$ are given in Table 3 and 5.


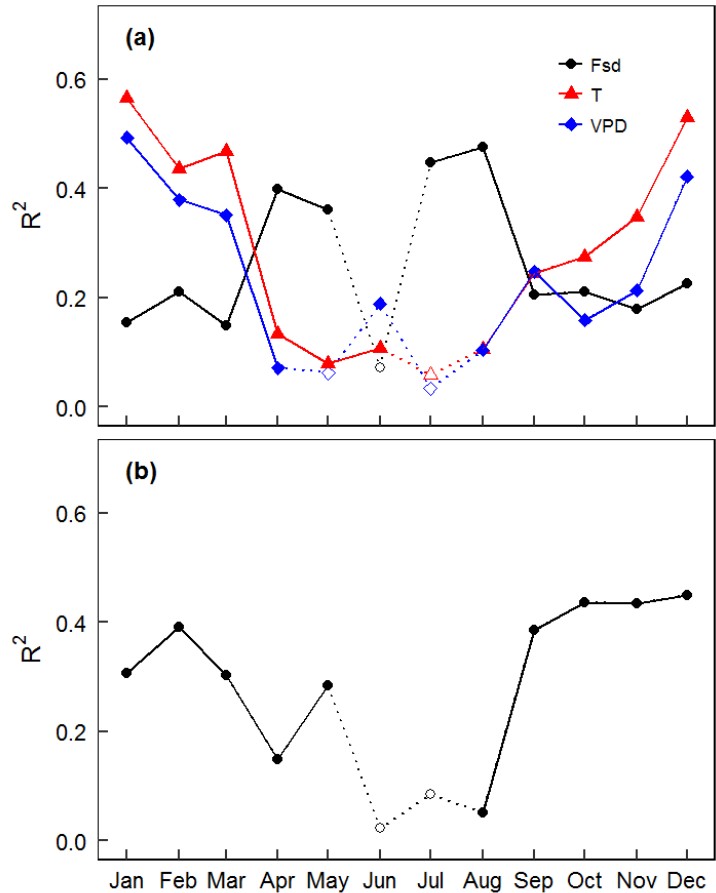

**Figure 6: Seasonal variability in environmental controls of day time and night time NEE; coefficients of determination ($R^2$) of (a) the rectangular hyperbolic light response curve (LRC) between daily means of midday NEE and incoming radiation (Fsd, black lines and circles), linear fits between residuals of LRC and air temperature (Ta, red lines and triangles) or vapour pressure deficit (VPD, blue lines and diamonds) and (b) the temperature response function after Lloyd and Taylor (1994) between daily means of u\* filtered night time NEE and air temperature (Ta, black lines and circles) per month pooled over three years; open symbols indicate non significant $R^2$; nr per month for (a) from Jan to Dec: 62, 74, 77, 76, 58, 42, 56, 60, 74, 75, 66, 72; nr per month for (b) from Jan to Dec: 55, 69, 63, 66, 52, 45, 41, 57, 63, 63, 52; 68**

**Table 1: Site and tower characteristics for the Wombat State Forest OzFlux-site**

| | |
|---|---|
| Location | 37° 25' S, 144° 05' E |
| Elevation a.s.l. (m) | 706 |
| Forest size (ha) | 70 000 |
| Tower height (m) | 30 |
| Canopy height (m) | 22^ |
| Canopy species | *Eucalyptus obliqua, E. rubida, E. radiata* |
| Understorey species | *Pteridium esculentum, Tetrarrhena juncea, Poa sieberiana, Lomandra* spp. |
| Mean annual air temperature (°C) | $11.0 \pm 0.1$ |
| Mean annual rainfall (114 yrs, mm)[#] | $879 \pm 18$ |
| *LAI* (leaf area index $m^2$ $m^{-2}$) | 1.81* |
| Tree density ($ha^{-1}$) | 1316* |
| Tree dbh (cm) | 18.6* |
| Litterfall ($g$ $m^{-2}$ $yr^{-1}$) | $1120 \pm 52$ |
| Soil type | acidic-mottled, dystrophic, yellow Dermosol |
| Soil depth (cm) | 50cm |
| pH | $4.83 \pm 0.02$ |
| Bulk density (0-10 cm, $kg$ $m^{-3}$) | $0.94 \pm 0.02$ |
| C/N | $30.9 \pm 0.5$ |
| Sand (%) | $45.4 \pm 1.8$ |
| Silt (%) | $27.9 \pm 1.9$ |
| Clay (%) | $26.7 \pm 0.4$ |

where applicable: mean of n = 3 ± 1SE; # BoM station Daylesford, 11 km N of study site), ^ Griebel et al. (2015), * Moore (2011)

**Table 2: Annual averages of incoming solar radiation (Fsd), air temperature (Ta), soil moisture content (SWC), vapour pressure deficit (VPD), annual sums of rainfall (P), net ecosystem productivity (NEE), ecosystem respiration (ER) and gross primary productivity (GPP) at the Wombat State Forest from 2010 to 2012; CV – coefficient of variation for inter-annual variation, inter-annual differences are indicated with *** (p<0.001), ** (p<0.01) or *ns* (not significant), letters indicate year to year differences**

| Year | Fsd ($W\ m^{-2}$) | Ta (ºC) | Ts (ºC) | SWC (v/v) | VPD (kPa) | P (mm) | NEE ($g\ C\ m^{-2}$) | ER ($g\ C\ m^{-2}$) | GPP ($g\ C\ m^{-2}$) |
|---|---|---|---|---|---|---|---|---|---|
| **2010**[a] | 174 | 11.0 | 12.1 | 0.25*a* | 0.36*a* | 1254 | -1046*a* | 1603*a* | 2649 |
| **2011** | 177 | 11.1 | 11.8 | 0.27*b* | 0.32*a* | 1070 | -1231*b* | 1534*a* | 2764 |
| **2012** | 182 | 11.1 | 11.8 | 0.24*a* | 0.47*b* | 872 | -1424*c* | 1346*b* | 2770 |
| **CV (%)** | 2.5 *ns* | 0.5 *ns* | 1.2 *ns* | 6.6 *** | 20.0 *** | 17.9 | 15.3 *** | 8.9 *** | 2.5 *ns* |

[a] includes extrapolated values until 21[st] of January, estimates without extrapolation, i.e. 344 days:
P (mm): 1229, NEE ($g\ C\ m^{-2}$): -958, ER ($g\ C\ m^{-2}$): 1476, GPP ($g\ C\ m^{-2}$): 2434

**Table 3: Parameters, standard errors and/or coefficient of determination ($R^2$) of (a) the rectangular hyperbolic light response curve (LRC) between daily means of midday NEE and incoming radiation (Fsd), (b) linear fits between residuals of LRC and air temperature (Ta) or vapour pressure deficit (VPD) and (c) linear fits between 2[nd] residuals (b) with Ta and soil water content (SWC) for subsets of data, α: initial slope of LRC and canopy light utilization efficiency ($\mu mol\ CO_2\ J^{-1}$), β: maximum NEE (i.e. uptake rate of the canopy) at light saturation ($\mu mol\ CO_2\ m^{-2}\ s^{-1}$); significance level: *** <0.001, ** <0.01, * <0.5, ns: not significant**

| data subset | (a) Fsd α (se) | | β (se) | | $R^2$ | (b) T $R^2$ | | VPD $R^2$ | | (c) SWC $R^2$ | | nr |
|---|---|---|---|---|---|---|---|---|---|---|---|---|
| **All data** | -0.14 0.01 *** | | -21.8 0.0 *** | | **0.25** | **0.18** *** | | **0.23** *** | | **0.05** *** | | 792 |
| **2010** | -0.11 0.02 *** | | -22.5 0.0 *** | | **0.28** | **0.19** *** | | **0.25** *** | | **0.05** *** | | 213 |
| **2011** | -0.13 0.02 *** | | -21.9 0.0 *** | | **0.27** | **0.21** *** | | **0.21** *** | | **0.10** *** | | 292 |
| **2012** | -0.17 0.02 *** | | -21.3 0.0 *** | | **0.22** | **0.15** *** | | **0.24** *** | | **0.04** *** | | 287 |

Table 4: Parameters, standard errors and/or coefficient of determination ($R^2$) of the rectangular hyperbolic light response curve (LRC) between daily means of midday NEE and incoming radiation (Fsd) for various Ta bins, VPD bins and SWC bins; α: initial slope of LRC and canopy light utilization efficiency (µmol $CO_2$ $J^{-1}$), β: maximum NEE (i.e. uptake rate of the canopy) at light saturation (µmol $CO_2$ $m^{-2}$ $s^{-1}$); significance level: *** <0.001, ns: not significant

| LRC | α (se) | | | β (se) | | | $R^2$ | nr |
|---|---|---|---|---|---|---|---|---|
| **Ta bins** | | | | | | | | |
| **<8** | -0.14 | 0.02 | *** | -23.6 | 0.0 | *** | **0.39** | 133 |
| **8-12** | -0.09 | 0.01 | *** | -29.3 | 0.0 | *** | **0.55** | 205 |
| **12-16** | -0.07 | 0.01 | *** | -30.4 | 0.0 | *** | **0.50** | 172 |
| **16-20** | -0.07 | 0.01 | *** | -27.9 | 0.0 | *** | **0.37** | 124 |
| **20-24** | -0.08 | 0.02 | *** | -20.7 | 0.0 | *** | **0.19** | 97 |
| **24-28** | -0.06 | 0.03 | ns | -20.5 | 0.1 | *** | 0.13 | 49 |
| **VPD bins** | | | | | | | | |
| **<0.4** | -0.10 | 0.01 | *** | -28.7 | 0.0 | *** | **0.46** | 324 |
| **0.4-0.8** | -0.07 | 0.01 | *** | -30.5 | 0.0 | *** | **0.44** | 232 |
| **0.8-1.2** | -0.05 | 0.01 | *** | -32.9 | 0.0 | *** | **0.47** | 103 |
| **1.2-1.6** | -0.62 | 1.39 | ns | -16.7 | 0.7 | *** | 0.00 | 68 |
| **1.6-2.0** | -0.16 | 0.25 | ns | -16.3 | 0.5 | *** | 0.02 | 25 |
| **2.0-2.4** | -0.06 | 0.05 | ns | -18.7 | 0.2 | *** | 0.09 | 26 |
| **2.4-2.8** | -0.05 | 0.03 | ns | -15.1 | 0.2 | *** | 0.47 | 7 |
| **2.8-3.2** | - | | | - | | | - | |
| **>3.2** | -0.08 | 0.27 | ns | -12.7 | 0.8 | *** | 0.05 | 4 |
| **SWC bins** | | | | | | | | |
| **<0.22** | -0.11 | 0.02 | *** | -22.35 | 0.0 | *** | **0.18** | 258 |
| **>=0.22** | -0.14 | 0.01 | *** | -21.88 | 0.0 | *** | **0.29** | 534 |

Table 5: Parameters, standard errors and/or coefficient of determination ($R^2$) of the temperature response function after Lloyd and Taylor (1994) between daily means of u* filtered night time NEE and air temperature (Ta) for subsets of data, $R_{ref}$: basal respiration rate at 10°C (µmol $CO_2$ $J^{-1}$), E: activation energy related parameter; significance level: *** <0.001, ** <0.01, * <0.5, ns: not significant

| LT | $R_{ref}$ (se) | | | E (se) | | | $R^2$ | nr |
|---|---|---|---|---|---|---|---|---|
| **All data** | 3.0 | 0.1 | *** | 310 | 16 | *** | **0.36** | 694 |
| **2010** | 3.6 | 0.2 | *** | 215 | 23 | *** | **0.31** | 214 |
| **2011** | 2.8 | 0.2 | *** | 405 | 27 | *** | **0.49** | 234 |
| **2012** | 2.8 | 0.2 | *** | 306 | 31 | *** | **0.30** | 246 |
| **SWC bins** | | | | | | | | |
| **0.1-<0.15** | 2.1 | 0.5 | *** | 349 | 94 | *** | **0.34** | 37 |
| **0.15-<0.20** | 3.0 | 0.3 | *** | 285 | 44 | *** | **0.27** | 138 |
| **0.20-<0.25** | 3.1 | 0.2 | *** | 339 | 34 | *** | **0.38** | 146 |
| **0.25-<0.30** | 3.4 | 0.2 | *** | 337 | 30 | *** | **0.38** | 200 |
| **>0.30** | 3.3 | 0.2 | *** | 527 | 40 | *** | **0.42** | 173 |