# Peer review of "Net ecosystem carbon exchange of a dry temperate eucalypt forest"

_Biogeosciences, 2016_

## Referee Comment (RC1) · Anonymous Referee #1 · 18 May 2016

Summary Statements:

Papers from the OzFlux network are needed to document carbon (and water) flux patterns for a host of ecosystem types that have heretofore been under-represented in global measurement networks. By sampling a drier temperate eucalyptus forest than has been studied in the past, this study brings unique observations of ecosystem scale carbon fluxes that provide a valuable expansion of the Australian and (semi-)global ecosystem flux measurement networks. The paper's major contribution and strength is to present estimates of daily, seasonal and annual NEE, GPP, and ER from multiple years of observation.

However, there are some significant problems with the methods used to estimate net ecosystem exchange from eddy covariance data, and also with the estimation of component gross carbon fluxes. Each concern is detailed below. Given the context of the special issue to which this paper has been submitted, it is an opportune, critical moment for ensuring high quality data and sound scientific standards for the data collection, data processing, and data analysis techniques being employed by this site's investigators and, by extension, the broader OzFlux community. Judging from what I see in this paper, and its author list that includes a number of leaders in the OzFlux network, it seems there is a need to underscore and reiterate the importance of thorough and careful implementation of EC methods. For example, the use of automated 30-minute flux calculations omits a series of important QA/QC filters and corrections that need to be applied to the high frequency (e.g. 10Hz data). This seems to have been overlooked in this work, possibly others. Implementing these post-processing steps is really not very difficult and is part of the industry standard for producing reliable datasets with the EC method. Similarly, the absence of a CO2 profile and associated storage correction to estimation of NEE from above-canopy Fco2 is quite disappointing given the stature of this forest. The EC system is also rather close to the canopy top. The footprint assessment and screening is insufficiently described. This and other problems need to be resolved or mitigated as much as possible before the paper is reconsidered for publication. They should also be brought into the routine methods used as part of OzFlux.

The paper also has some major problems with its methods of data analysis. The methods of analysis are imprecise, misleading, and even circular in a number of places. Corresponding interpretations are significantly flawed, and most of the conclusions are unsupported. Analyses do not offer insight into the functional controls or functional responses of the ecosystem to environmental conditions. Sadly the paper does not present light response curves and parameters, temperature response parameters, sensitivity to VPD, or other typical measures and metrics that one would expect to see and that would lend themselves to broader comparison to other sites and also provide a lense from which to address questions about what processes and conditions drive variations in the measured fluxes. The paper also fails to deliver on its objective of attributing seasonal and interannual variability in carbon fluxes to environmental drivers. The design and structure of analyses that intende to do this are unsound. The underlying dataset may have the potential to deliver useful insights about the drivers of seasonal and interannual variability of interest but the method of analysis would need to be substantially altered. Suggestions are offered below. Though I would like to be able to recommend that this section (e.g. Figs 5 and 6, Tables 2 and 3) simply be dropped from the paper, that would leave very little in the paper beyond a simple presentation of the basic data (such as in Figures 1 through 4). Reporting rates of ecosystem fluxes from a single site without a cogent analysis of the underlying drivers or the functional responses is arguably below the bar for publication in this journal. I have no doubt that the dataset can be used to produce a more insightful and valuable contribution but unfortunately the analysis and paper does not do so in its present state.

In sum, I would suggest that the paper be completely reworked before further consideration, with attention to the core data stream and to the analyses presented and corresponding interpretations.

Main Concerns:

(1) The Study's Measurement Techniques and Data Post-Processing: The data processing is inadequate and even the core measurement technique is suspect.

(1a) Measurement of turbulent fluxes at only 3 meters above a 30 meter canopy puts these observations barely into the roughness sublayer where the near-field effects of scalar sources and sinks can have a major influence. In other words, it is unclear that these measurements can provide a reliable sample of well-mixed turbulent fluxes.

(1b) Measurement atop a ridge with sloping terrain in prevailing wind directions could significantly bias flux estimates and one cannot help but wonder what role advection fluxes may play in the very strong carbon sink that is reported for the site.

(1c) It is well known that above canopy turbulent exchange cannot be assumed to

represent net ecosystem exchange over tall canopies because of the build-up of $CO_2$ during conditions of low turbulence, the subsequent ventilation of $CO_2$ when more turbulent conditions resume, as well as depletion of within-canopy $CO_2$ by uptake in the canopy but that may not be detected as prompt above-canopy flux. Therefore, estimates of NEE over tall canopies requires measurements of the vertical profile of $CO_2$ to estimate the storage flux in addition to turbulent above canopy flux. Unfortunately, these measurements do not appear to have been included in the present deployment. There is no mention of data post-processing techniques that are used to mitigate this problem and the effects can be quite significant. It is a problem not only for NEE but also the gross fluxes inferred from NEE separation. This, too, calls into question the validity of the data record and the findings regarding large and persistent carbon dioxide uptake at the site.

(1d) It would seem that $CO_2$ emissions from the diesel generator could appear as an erroneous $CO_2$ source from the ecosystem. The text simply states that the generator is 'remote' but how remote can it be given concerns of power loss over long cables. How this concern is mitigated should be clarified.

(1e) The explanation of footprint modeling and associated screening as well as screening based on wind direction so that "fluxes were constrained to the same forest type and dominant tree species" is inadequately described. Citing the Griebel et al. 2016 study is incomplete. The method should be explained more fully here.

(1f) Measurement of soil moisture at 5 cm depth is unlikely to represent the soil water status relevant to 25 m tall trees. Measurement at a single point is also unlikely to be adequately representative given the typically large spatial variability of soil water content.

(1g) Measurement of rainfall with a rain gage placed 1 m above the ground where trees are 25 m tall is sure to undersample the real rainfall rate above the canopy.

(1h) Data post-processing is inadequate. It is customary and important to perform postprocessing on the high frequency (here 10Hz) data rather than to rely on automated online software to calculate half-hour averaged fluxes. Unfortunately this was not done for the present work. There does not appear to be any check for non-stationarity and detrending over the 30-minute interval. There does not appear to be any despiking of raw 1/10th of a second data, certainly not the recommended method of despiking based on a moving assessment of instantaneous data relative to the standard deviation in a time-local window of data. There cannot possibly be a co-spectral correction for issues of instrument separation or frequency attenuation of measurements, for specific conditions of stability, wind direction, turbulence intensity. There was only a simple 2D coordinate rotation rather than a full planar coordinate rotation such as with the Wilczak method. All of these elements are missing and are standard requirements for the production of reliable eddy covariance flux estimates.

(1i) NEE separation was done four ways but then the results of only a single method were selected without adequate justification. It would seem more appropriate to present results from all four methods as a source of methodological uncertainty.

(1j) A description of the method of NEE separation is needed. For example, what was the temporal window over which data were used to assess the relationship between ER (from selected Fc measurements) and environmental variables (principally temperature but also some others)?

(2) The Study's Methods of Data Analysis are Significantly Flawed.

(2a) Did analysis of the drivers of temporal variability rely on gap-filled data or only on trusted direct observations? If gap-filled data were used then all of these analyses are circular.

(2b) It is logically unsound to ask if ER relates to temperature when ER is determined based on a relationship between Fc and temperature. It is circular to use temperature to model ER over time and then to assess the importance of temperature for determining temporal variability in ER.
(2c) The analysis that alleges to diagnose the relative importance of different environmental variables in driving seasonal variability is significantly flawed. The authors examine which environmental variable best explains variability in 30-minute 'daytime' data for each month of the year and for each flux separately. This analyzes sources of diel variation, or hourly variation, not seasonal variation. Day-time appears to be defined here as half-hours when shortwave radiation was greater than 10 W m-2. Not surprisingly, sunlight turns out to be an important variable for GPP for all months of the year. This does NOT indicate that solar radiation is the dominant factor governing seasonal variability in carbon fluxes at the site. To assess that you would need to do something like use daily data, or mean daylight values for each day, and then study the full year to explore relationships to environmental variables. You would need to consider LAI dynamics as well. Of course there will likely be collinearity in these daily values, but it at least has the right time scale to answer your question of attributing seasonal variability to drivers. It is also not surprising the temperature turns out to be important for ER variability at the within-month time scale because temperature is one of the predictor variables used to model ER. Again, this is circular.

(2d) The analysis that intends to identify the relative importance of different drivers of interannual variability is also severely flawed. What the authors examine, really, is if the relative importance of drivers that determine 30-minute fluxes over the whole year differs between years. Not only does this alias diel variation into the analysis as noted above (2c) but it also conflates all seasonal variability as well. What should be done instead would be to examine daily (or monthly) anomalies relative to the across-year mean for that day (or mean monthly values). Or, you could study if the functional response of fluxes to PAR, air temperature, VPD, or soil moisture differed across years, stratifying to control for the effects of the other variables. Simultaneously you would want to study if the environmental conditions varied meaningfully across years, partly examined earlier in the paper but not linked to implications for fluxes. What might be best would be a real attribution exercise in which you use the 2010 functional response to environmental conditions to estimate what the 2011 and 2012 fluxes would have

been with that functional response surface and then compare that synthetic result to the real measured case. You could do the same for the other years. Similarly, you could use the environmental conditions of 2010 combined with the functional response of 2011 (and then 2012) to estimate the effects of any drift in the functional responses alone by, again, comparing to the real case. This would allow you to truly assess the drivers of IAV in fluxes as being due to variation in environmental conditions, variation in flux responses to environmental conditions, or both. Unfortunately what has been done is simply not testing anything meaningful about IAV in fluxes.

(2e) For the above reasons, much of the discussion section on environmental drivers of CO2 fluxes includes misinterpretations.

(3) Discussion is Overly Confident and Conclusions are Unsupported:

(3a) L 296: Comparison to these other forest types is valuable but the narrative is overly confident here. You cannot state that leaf longevity explains the differences in GPP rates between these various forests. The findings presented in this paper do not substantiate that supposition or conjecture.

(3b) L304: Delayed spring increase in ER relative to that for GPP may be partly due to soil respiration but it could still be partly linked to temperature control on plant respiration, no? Soil respiration is not shown in the present study and so this remains supposition.

(3c) L335: This site's very large carbon sink (NEE of around -1,000 g C m-2 y-1) is surprising and noteworthy. Could it be related to secondary recovery and the site's 25 year old stand age? What is the disturbance legacy for this site?

(3d) L383: It is not much of a finding to discover that seasonality is different in a dry, warm (winter free) temperate eucalypt site compared to temperate coniferous and deciduous forest sites with a strong seasonality in climate with sustained winter freezing. Furthermore, this study did not really show the difference in seasonality explicitly. Also,

it is stated that this alleged difference in seasonality is due to the opportunistic response of eucalypt forests. This is not supported by any analysis and it is not even clear what is meant by "opportunistic response". A simpler explanation is that they in a different climate setting.

(3e) L 385: This study did demonstrate "that seasonal and inter-annual variability in carbon uptake were not limited by temperature but predominantly driven by radiation". The study also did not demonstrate that "carbon loss from the forest was dominated and overall ecosystem carbon exchange dynamics were not water limited due to the high rainfall" and this sentence has a hanging statement (its first part about carbon loss).

(3f) L 387: Nothing presented in the paper quantitatively supports the statement that "temperate eucalypt forests represent a unique forest type and should be considered separately in future classifications of ecoystems...". To demonstrate this you would need to show that these forests have a different functional response to environmental conditions than other forest types. This has not been shown and would require a synthesis analysis, not simply data from a single site. It is entirely possible that if you were to control for site and time specific conditions of LAI, PAR, VPD, Tair, and soil moisture you might find that these forests behave similarly to others. This has not been tested in the present study.

(3g) L 389: Could drop the last sentence. It doesn't seem to add much of substance and I'd argue it does not really need to be said. It sounds a bit like a proposal statement or a sales pitch.

(4) More Minor Presentation Issues.

(4a) Figure 2: it would be better to use a line chart for panel a rather than a bar chart.

(4b) Figure 2: it would be helpful to show midday VPD either in addition to or instead of mean daily VPD.

---

## Referee Comment (RC2) · Anonymous Referee #2 · 7 Jun 2016

The carbon exchange of Eucalypt forests, which have the rare characteristic of being broadleaf evergreen, is still not well understood. This study presents the first three years of eddy covariance measurements of carbon exchange in a dry Eucalypt forest in Australia. The forest is found to be a very large sink of carbon at more than a 1000 g C m-2 yr-1 due to yearlong GPP and a rather low ER. The main environmental drivers were assessed using a random forest analysis and the main drivers were found to be solar radiation and air temperature.

The paper brings forward some very important numbers in terms of carbon budgets for this dry temperate eucalypt forest that were missing from global flux measurements, and reveals among the largest CO2 sequestration for any ecosystem studied thus far. Such numbers, as well as how the fluxes vary over time (diurnal, seasonal, interannual), are highly needed to better understand how different ecosystems vary in their

growth dynamics and ecophysiology. However, I have a series of serious concerns which I suggest need to be addressed for the paper to be considered for publication.

Main concerns:

1) On line 105, the forest height is said to be 21-27 m, while in Table 1 it is 25-27 m. Which one is it? How uniform is the distribution of tree heights in the forest? The reason why this matters is that the eddy covariance (EC) system is at a height of 30 m, thus only 3 m above the tallest trees. How turbulent are the winds at 30 m? Did you assess whether the EC system is high enough to provide reliable fluxes or if it is located too close to the sources/sinks of fluxes and thus not well-mixed? For e.g., the wet eucalypt forest in the paper from van Gorsel et al. 2013 has a tree height of 40 m and the EC system is at a height of 70 m. Why are the measurements made so close to the canopy at this site? Please show that the EC measurements are reliable at this site.

2) The flux tower is located on a ridge surrounded by some gently sloping gullies. Does the ridge affect the wind directions? Did you investigate if it leads to some advection or decoupling within the forest canopy? This is especially important for your dry eucalypt forest, as the numbers you report are extremely large for NEE while ER is low. This suggests a potential for advection within the canopy, which needs to be addressed for these numbers to be considered reliable. It is puzzling that a forest with such a low LAI sequesters so much carbon. Please consider the possible influence of advection on your fluxes. Also, please put more emphasis on the reasons why ER is so low because for the reader at the moment this is not well explained and raise skepticism on advection. How much organic matter/litter is on the soil surface? Do the eucalypt leaves ever fall? Do the trees themselves respire less than other type of trees?

3) The u* threshold is unusually high at 0.56 to 0.69 m s-1. This is above most u*-thresholds reported in the literature (see for e.g. Papale et al. 2006 Biogeoscience Fig.1) Why is it so high? Greater details need to be given on how this threshold was

determined. Is there really an inflection point around those values? In my experience, when the common methods determine such a high threshold, it is when they do not apply i.e. there is no inflection point because it is a site with advection or with some other abnormality making the u*-threshold determination not applicable.

4) As a result of the very high u*-threshold, only a low number of actually measured data is kept for the three years (37%, 47%, 47%). This means that the outputted annual budgets depend mostly on the gap filling and thus, represent more modelled numbers rather than measurements. Sometimes, depending on the data quality, it can be better to use less ideal, measured data than modelled data. How different would the cumulative numbers be if for e.g. 60-70% of the measured data is kept?

5) On lines 123-125, the writing suggests that only the 30 min fluxes were stored, is that correct? If not please add more details. If it is correct, this seems like it would bias your fluxes because you cannot quality filter the raw data before flux calculations so any spikes in the raw fluxes is reflected in the 30 min fluxes. Although this cannot be changed now, it would be needed in the future to store the raw fluxes as well. In addition, did you estimate the $CO_2$ storage in the canopy?

6) The authors use the random forest algorithm to determine the environmental drivers for NEE, GPP and ER. This analysis shows some serious circular argumentation. GPP is partitioned using solar radiation, thus it is beyond expected that there will be a strong relationship there. Same for ER and air temperature. Instead, I think what would help is to look at functional relationships between GPP and solar radiation, and between ER and air/soil temperature. This would provide the reader with a better understanding of how the dry eucalypt forest behaves compare to other ecosystems, as the numbers from photosynthetic capacities and such could be directly compared to those of other ecosystems. This would also add to the discussion when compared with other ecosystems to help us understand what is so different about this forest so that it sequesters so much CO2. . . At the moment the paper does not deliver clearly what makes this ecosystem so different. Improving this analysis as well as providing functional relationships figures would help to address objective 2.

Minor comments:

Lines 51 to 57: there are four "while" in those 7 lines, including two in one sentence. Please reduce the number of "while".

Line 66: comma wrongly placed

Line 77: comma missing between abundant and there

Line 105: describe tree height better (how uniform and such) and add LAI in text.

Line 114: Is it really a mean of 114 years??

Lines 130-131: Having one soil temperature, one soil moisture and one soil heat flux plate measurement is not sufficient to characterise soil due to its spatial variability. You need replicates horizontally and vertical profiles within the soil. On line 131, SWC is at 5 cm depth then on line 149 you talk about 10 cm depth. Please clarify the soil measurement depths.

Line 144: There is an additional space before Contiuum.

Lines 145-146: Greater details need to be given on the quality filtering. This is not clear. What range checks? What thresholds for spikes? What thresholds for outliers? etc. . .

Line 169: Is that how LAI was derived? Are there any ground measurements of LAI to validate the MODIS data?

Section 2.3.3: it is confusing which method you used in the end for partitioning. Please give the equations used for partitioning and show a figure comparing the different approaches if you mention you compared different methods in text. Also, show a comparison with the soil respiration measurements that you mention on line 185.

Line 301: temperate evergreen coniferous forests (add evergreen). It is not because

they are coniferous but rather because they are evergreen.

Figure 2(a) is it NEE or –NEE (why is there a minus in the caption?) Same for Figure 3 and 4. Also, please describe what you display better, for e.g. in figure 3, what are the shaded lines at the back, the actual daily totals? Typically NEE is what is measured by EC, and sometimes people convert it to NEP=-NEE. In your case, you do not need the minus there because you display NEE.
* * *

---

## Author Comment (AC1) · 3 Sep 2016

*Authors' response* to reviewer comments on "Net ecosystem carbon exchange of a dry temperate eucalypt forest" *by Hinko-Najera et al.* (bg-2016-192)

We would like to thank both reviewers for their time spent reading the manuscript, their constructive comments and valuable suggestions on changes and/or additions particularly regarding methodology and data analysis that, we think, will strongly improve our manuscript. We have addressed all comments and issues raised by the reviewers, propose changes to the manuscript accordingly and provide following detailed responses to reviewer comments below. For facilitation we have copied the comments of both reviewers (black text) and our responses are outlined in blue.

**Relevance of the study and major issues raised by both reviewers:**

Some of the major concerns of both reviewers were similar and thus are outlined in summary points as follows:

- **Relevance of the study:**

**R1:** By sampling a drier temperate eucalyptus forest than has been studied in the past, this study brings unique observations of ecosystem scale carbon fluxes that provide a valuable expansion of the Australian and (semi-)global ecosystem flux measurement networks. The paper's major contribution and strength is to present estimates of daily, seasonal and annual NEE, GPP, and ER from multiple years of observation.

**R2:** The paper brings forward some very important numbers in terms of carbon budgets for this dry temperate eucalypt forest that were missing from global flux measurements, and reveals among the largest $CO_2$ sequestration for any ecosystem studied thus far. Such numbers, as well as how the fluxes vary over time (diurnal, seasonal, inter-annual), are highly needed to better understand how different ecosystems vary in their growth dynamics and ecophysiology.

Response: Both reviewers highlight the importance of the study and agree that it is addressing an important knowledge and data gap that is relevant on a global scale to improve our process understanding.

- **Measurement or tree height (comments R1.1a, R2.1):**

**R1.1a:** Measurement of turbulent fluxes at only 3 meters above a 30 meter canopy puts these observations barely into the roughness sublayer where the near-field effects of scalar sources and sinks can have a major influence. In other words, it is unclear that these measurements can provide a reliable sample of well-mixed turbulent fluxes.

**R2.1:** On line 105, the forest height is said to be 21-27 m, while in Table 1 it is 25-27 m. Which one is it? How uniform is the distribution of tree heights in the forest? The reason why this matters is that the eddy covariance (EC) system is at a height of 30 m, thus only 3 m above the tallest trees. How turbulent are the winds at 30 m? Did you assess whether the EC system is high enough to provide reliable fluxes or if it is located too close to the sources/sinks of fluxes and thus not well-mixed? For e.g., the wet eucalypt forest in the paper from van Gorsel et al. 2013 has a tree height of 40 m and the EC system is at a height of 70 m. Why are the measurements made so close to the canopy at this site? Please show that the EC measurements are reliable at this site.

Response: Both reviewers have expressed concern that the eddy covariance (EC) measurements are not reliable given the stated range of tree heights and the height of the EC measurements. We acknowledge the concern of the reviewers and will address these by 1) updating the canopy height given in the existing manuscript with a more accurate value that has been measured using ground-based Lidar, 2) clarifying the existing description of the EC measurements, and 3) adding text that directly addresses the issues surrounding EC measurements within the roughness sub-layer (RSL).

In addressing this issue, we also make the following 3 points:

1) The stated tree height of the forest in both the text and in Table 1 was based on an initial approximation and has since been replaced with more accurately measured figures. We will correct the text and Table 1 accordingly in the revised version of the manuscript. The actual canopy height is 22 m and based on terrestrial Lidar measurements as presented in a recent publication about the site (Griebel et al., 2015). The revised canopy height increases the height of the EC instruments above the canopy to 8 m (emergent trees will reduce this but their contribution to the total LAI is estimated to be less than 5%). This is still within the RSL using the definition of RSL depth as between 2 and 5 times the canopy height (Katul et al., 1999). We discuss the implications of this on the measurements in the next paragraph.

2) Katul et al. (1999) give a detailed report of an experiment performed over the Duke Forest, Durham, North Carolina, USA. The objective of the experiment was to explore the effect of taking EC measurements within the RSL by comparing data from 6 EC systems mounted at 15.5 m over a 14 m canopy (managed loblolly pine). This is 1.5 m above the canopy compared to the 8 m height difference at our study site (based on a more accurate measurement of canopy height, see above). Given the greater height difference between the EC instruments and the canopy at our study site, we can reasonably expect the results in Katul et al. (1999) to represent a worst case for our study site. In the context of the measurements taken at our study site, the important conclusion Katul et al. (1999) state is that the observed site-to-site differences in $CO_2$ (and $H_2O$) fluxes of 20% may be a result of taking measurements within the RSL and may not be attributable to site-to-site differences in phenology or function. Translating this result to a single site implies that we may expect ±10% uncertainty (based on the 20% range quoted in Katul et al. 1999) about a mean value (half the observed range from highest to lowest) due to uncertainty associated with the measurements being taken within the RSL. Again, we stress that the EC instruments at our study site are more than 5 times higher above the canopy than those used in Katul et al. (1999). These results suggest that errors due to the location of EC instruments in the RSL are of the order of 10% at our study site.

3) Barr et al. (2013) used data from 38 sites from Canada and the USA in their investigation of the use of the change point detection (CPD) method for determining the u* threshold. 10 of these sites have canopy heights of 20 m or greater. Of these 10, 7 have their EC instruments located 10 m (half canopy height) or less above the canopy. Location of EC instruments in the RSL over tall canopies is not uncommon. We do not suggest by this that it is not a problem, the effect of taking EC measurements in the RSL certainly needs more study, but that our study site is not alone in having this problem.

- **The use of 30 min averaged data (comments R1.1h, R2.5):**

**R1.1h:** Data post-processing is inadequate. It is customary and important to perform post processing on the high frequency (here 10Hz) data rather than to rely on automated online software to calculate half-hour averaged fluxes. Unfortunately this was not done for the present work. There does not appear to be any check for non-stationarity and detrending over the 30-minute interval. There does not appear to be any despiking of raw 1/10th of a second data, certainly not the recommended method of despiking based on a moving assessment of instantaneous data relative to the standard deviation in a time-local window of data. There

cannot possibly be a co-spectral correction for issues of instrument separation or frequency attenuation of measurements, for specific conditions of stability, wind direction, turbulence intensity. There was only a simple 2D coordinate rotation rather than a full planar coordinate rotation such as with the Wilczak method. All of these elements are missing and are standard requirements for the production of reliable eddy covariance flux estimates.

**R2.5:** On lines 123-125, the writing suggests that only the 30 min fluxes were stored, is that correct? If not please add more details. If it is correct, this seems like it would bias your fluxes because you cannot quality filter the raw data before flux calculations so any spikes in the raw fluxes is reflected in the 30 min fluxes. Although this cannot be changed now, it would be needed in the future to store the raw fluxes as well.

Response: Both reviewers have expressed concerns about the method used to process data presented in this manuscript. The EC data was processed using "OzFluxQC" and gap-filled using DINGO. "OzFluxQC" is described in detail by Isaac et al. (2016), an already accepted paper with minor revisions in this Special Issue. DINGO is described in detail by Beringer et al. (2016) and also accepted with minor revisions in this Special Issue. However, the reviewers are correct to question the data processing where there is uncertainty over the method used.

Our response to these concerns from the reviewers has 2 parts. First, we will describe the extra work we intend to do to address the reviewers' primary concern (fluxes not calculated from the 10 Hz data in post-processing). Second, we will address some of the related points raised by the reviewers.

We will process the available 10 Hz data from our study site using EddyPro and will include these results in a revised version of the manuscript (note that Isaac et al. (2016) demonstrate the equivalence of calculating the fluxes from the covariances output by the data logger as done by OzFluxQC and calculating them from the 10 Hz data using EddyPro). As part of this process, we will compare results obtained from EddyPro using the 2D rotation method (used when OzFluxQC calculates the fluxes from the covariances) and using the planar fit method to establish if the different methods yield significantly different results at this site. We will also modify the section that describes the data processing in a revised version of the manuscript once the re-processing is complete to clarify the data processing methods and to include, where appropriate, references to Isaac et al. (2016) and Beringer et al. (2016) in this

Special Issue. Based on previous experience and on the work presented in Isaac et al (2016) we expect differences between the results to be small and unlikely to change the major conclusions of our manuscript.

To the related points raised by the reviewers:

1) R1: *"... to rely on automated online software to calculate half-hour averaged fluxes."*: The fluxes presented in the manuscript were not calculated by automated online software but are calculated by "OzFluxQC" during post-processing from the 30 minute average covariances output by the data logger after applying quality control checks and coordinate rotation to the covariances. This is mathematically equivalent to processing the 10 Hz data using EddyPro (or similar) with the exception of de-spiking and rejection of non-stationary time periods which we address below.

2) R1: *"There does not appear to be any despiking ..."*: The reviewer is correct that calculating the fluxes from the 30 minute covariances does not allow despiking in the traditional sense employed in EddyPro and other similar packages. However, the data logging methods used by OzFlux allow the sonic anemometer and IRGA diagnostic information to be recorded at the sampling frequency (10 Hz). The data logger program uses this diagnostic information to accept or reject each individual 10 Hz sample prior to the calculation of the covariances at the end of each 30 minute averaging period. To the extent that spikes in the sonic or IRGA outputs are detected by the instruments and flagged by the instrument's diagnostic information, this process will result in the removal of data spikes from the covariance calculation.

3) R1: *"There does not appear to be any check for non-stationarity and detrending ..."*: The data processing adopted by OzFlux does not apply any detrending. This is the approach recommended in Kaimal and Finnigan (1994). The reviewer is correct that the non-stationarity checks used by EddyPro and similar programs cannot be done on the 30 minute data. However, we believe that the quality control checks described in Isaac et al. (2016) act to reduce the effect of this on the final data set. In particular, we have found during processing at another site (Cumberland Plains, Alexis Renchon, personal communication) that non-stationary conditions are often associated with low u* conditions and are rejected when a u* filter is applied to the data. As indicated above, however, we will address this point by calculating fluxes directly from the 10 Hz data using EddyPro with the appropriate stationarity checks applied.

4) R1: *"There cannot possibly be a co-spectral correction ..."*: Correction of half-hourly

fluxes for path averaging, instrument separation and frequency response is done in OzFluxQC using the method of Massman (2000). However, we agree with the reviewer that this method lacks some of the features of other correction algorithms. We will address this point by calculating fluxes directly from the 10 Hz data using EddyPro with the appropriate frequency corrections applied.

5) R1: *"There was only a simple 2D coordinate rotation ..."*: We will address this point by calculating fluxes directly from the 10 Hz data using EddyPro with the planar fit option chosen.

6) R2: We believe that in addressing the concerns of reviewer 1 as outlined above we will also address the concerns about data processing raised by reviewer 2.

- **Possibility of advection (comments R1.1b, R2.2):**

**R1.1b:** Measurement atop a ridge with sloping terrain in prevailing wind directions could significantly bias flux estimates and one cannot help but wonder what role advection fluxes may play in the very strong carbon sink that is reported for the site.

**R2.2:** The flux tower is located on a ridge surrounded by some gently sloping gullies. Does the ridge affect the wind directions? Did you investigate if it leads to some advection or decoupling within the forest canopy? This is especially important for your dry eucalypt forest, as the numbers you report are extremely large for NEE while ER is low. This suggests a potential for advection within the canopy, which needs to be addressed for these numbers to be considered reliable. It is puzzling that a forest with such a low LAI sequesters so much carbon. Please consider the possible influence of advection on your fluxes. Also, please put more emphasis on the reasons why ER is so low because for the reader at the moment this is not well explained and raise scepticism on advection. How much organic matter/litter is on the soil surface? Do the eucalypt leaves ever fall? Do the trees themselves respire less than other type of trees?

Response: Both reviewers highlighted and made important comments regarding the issue of advection. The reviewers are correct that we cannot exclude the influence of advection fluxes in our forest study site and we will acknowledge this fact in the discussion and emphasise the possibility/probability of advection in conjunction with annual estimates in a revised version of the manuscript.

We have observed low night time fluxes of NEE, hence ER, which indicated a decoupling within the forest canopy and thus advection fluxes during night time. We tried to account for this by using a yearly u* thresholds determined via change point detection (Barr et al., 2013) and by using only flux data from the first 3 hours after sunset (Beringer et al., 2016; van Gorsel et al., 2007) to estimate/ calculate ER. As outlined briefly in the manuscript and in more detail further below, this data selection yielded the highest estimates of ER. We also will provide further possible explanations of low ER estimates. Leaf litter fall generally is highest during summer months concurrent to shoot growth and in agreement with highest GPP and ER estimates within a year.

We like to emphasize that we present as a first results of ecosystem carbon fluxes and their seasonality in a dry temperate eucalypt forest. The seasonal behaviour is unlikely to change or be affected by an overestimation of NEE which rather is a systematic error than a random error. However, we will highlight the need to further investigate/ account for advection in the future in the discussion. Moreover, preliminary results from an ongoing study to validate more recent annual NEE estimates with a biomass inventory (biomass increment and carbon content) indicate that the NEE estimates from the EC measurements are systematically 30% greater than the NPP of the tree biomass from the inventory methods (Bennett, Bennett & Arndt, unpublished). It is a reasonable assumption that tree NPP contributes the bulk of NEE and as such the inventory data would be a confirmation that the flux tower is underestimating ER, resulting in a greater NEE. However, the underestimation was systematic and very similar in each of the measurement years. We will include this in the discussion.

- **Storage term/ profile data (comments R1.1c, R2.5):**

**R1.1c:** It is well known that above canopy turbulent exchange cannot be assumed to represent net ecosystem exchange over tall canopies because of the build-up of $CO_2$ during conditions of low turbulence, the subsequent ventilation of $CO_2$ when more turbulent conditions resume, as well as depletion of within-canopy $CO_2$ by uptake in the canopy but that may not be detected as prompt above-canopy flux. Therefore, estimates of NEE over tall canopies requires measurements of the vertical profile of $CO_2$ to estimate the storage flux in addition to turbulent above canopy flux. Unfortunately, these measurements do not appear to have been included in the present deployment. There is no mention of data post-processing techniques that are used to mitigate this problem and the effects can be quite significant. It is a problem not only for NEE but also the gross fluxes inferred from NEE separation. This, too,

calls into question the validity of the data record and the findings regarding large and persistent carbon dioxide uptake at the site.

**R2.5:** In addition, did you estimate the $CO_2$ storage in the canopy?

Response: The reviewers made valuable suggestions regarding profile measurements and we agree with the reviewers that this needs to be included in the manuscript. A profile system similar to that described in McHugh et al. (2016) in the same Special Issue was installed at our study site in early February 2012. Hence, within the scope of this manuscript, we have 8 months (February to October 2012) worth of profile data as we experienced technical problems with the IRGA in November 2012 which lasted until the end of 2012. We have analysed/calculated the storage term following McHugh et al. (2016) and Finnigan (2006). However, the addition of the storage term only marginally changed the magnitude of NEE in 2012. Nonetheless, we will include the profile data, i.e. change in storage term, for 2012 and re-do the analysis in the revised version of the manuscript and will discuss the possible reasons for the small effect the storage term had on NEE. Furthermore, the "OzFluxQC" procedure provides the option of a single point calculation of the storage term. We will test if this single point storage term is comparable to the available profile derived storage term in 2012 which would allow us to account for a change in storage term for the years where no profile measurements were available.

- **Analysis of environmental drivers (R1.2a-e, R2.6)**

**R1.2:** The Study's Methods of Data Analysis are Significantly Flawed.

**R1.2a:** Did analysis of the drivers of temporal variability rely on gap-filled data or only on trusted direct observations? If gap-filled data were used then all of these analyses are circular.

**R1.2b:** It is logically unsound to ask if ER relates to temperature when ER is determined based on a relationship between Fc and temperature. It is circular to use temperature to model ER over time and then to assess the importance of temperature for determining temporal variability in ER.

**R1.2c:** The analysis that alleges to diagnose the relative importance of different environmental variables in driving seasonal variability is significantly flawed. The authors examine which environmental variable best explains variability in 30-minute 'daytime' data for each month of the year and for each flux separately. This analyzes sources of diel variation, or hourly variation, not seasonal variation. Day-time appears to be defined here as half-hours when shortwave radiation was greater than 10 W m-2. Not surprisingly, sunlight

turns out to be an important variable for GPP for all months of the year. This does NOT indicate that solar radiation is the dominant factor governing seasonal variability in carbon fluxes at the site. To assess that you would need to do something like use daily data, or mean daylight values for each day, and then study the full year to explore relationships to environmental variables. You would need to consider LAI dynamics as well. Of course there will likely be collinearity in these daily values, but it at least has the right time scale to answer your question of attributing seasonal variability to drivers. It is also not surprising the temperature turns out to be important for ER variability at the within-month time scale because temperature is one of the predictor variables used to model ER. Again, this is circular.

**R1.2d:** The analysis that intends to identify the relative importance of different drivers of interannual variability is also severely flawed. What the authors examine, really, is if the relative importance of drivers that determine 30-minute fluxes over the whole year differs between years. Not only does this alias diel variation into the analysis as noted above (2c) but it also conflates all seasonal variability as well. What should be done instead would be to examine daily (or monthly) anomalies relative to the acrossyear mean for that day (or mean monthly values). Or, you could study if the functional response of fluxes to PAR, air temperature, VPD, or soil moisture differed across years, stratifying to control for the effects of the other variables. Simultaneously you would want to study if the environmental conditions varied meaningfully across years, partly examined earlier in the paper but not linked to implications for fluxes. What might be best would be a real attribution exercise in which you use the 2010 functional response to environmental conditions to estimate what the 2011 and 2012 fluxes would have been with that functional response surface and then compare that synthetic result to the real measured case. You could do the same for the other years. Similarly, you could use the environmental conditions of 2010 combined with the functional response of 2011 (and then 2012) to estimate the effects of any drift in the functional responses alone by, again, comparing to the real case. This would allow you to truly assess the drivers of IAV in fluxes as being due to variation in environmental conditions, variation in flux responses to environmental conditions, or both. Unfortunately what has been done is simply not testing anything meaningful about IAV in fluxes.

**R1.2e:** For the above reasons, much of the discussion section on environmental drivers of CO2 fluxes includes misinterpretations.

**R2.6:** The authors use the random forest algorithm to determine the environmental drivers for NEE, GPP and ER. This analysis shows some serious circular argumentation. GPP is

partitioned using solar radiation, thus it is beyond expected that there will be a strong relationship there. Same for ER and air temperature. Instead, I think what would help is to look at functional relationships between GPP and solar radiation, and between ER and air/soil temperature. This would provide the reader with a better understanding of how the dry eucalypt forest behaves compare to other ecosystems, as the numbers from photosynthetic capacities and such could be directly compared to those of other ecosystems. This would also add to the discussion when compared with other ecosystems to help us understand what is so different about this forest so that it sequesters so much CO2. At the moment the paper does not deliver clearly what makes this ecosystem so different. Improving this analysis as well as providing functional relationships figures would help to address objective 2.

Response:

1) Much of the concerns raised by reviewer 1 referred to the use of half-hourly data to assess seasonal and inter-annual variability in environmental drivers. We would like to clarify that we used daily averages of gap-filled ecosystem carbon fluxes binned per month or year for the random forest approach to assess seasonal and inter-annual variability in environmental drivers. However, we acknowledge that using daily averages was not clearly enough stated throughout the manuscript and in figure captions and will do so in a revised version of the manuscript.

2) We further acknowledge that using gap-filled data for the random forest approach is circular and thus potentially erroneous. We initially thought using valid observations with gaps could introduce a greater bias to such an analysis as gaps might occur predominantly for a particular time of day or year. However, we re-analyse the data with valid observations only in a revised version of the manuscript. Therefore we will use mean midday (11:00-13:00) values of measured NEE representing carbon uptake and the mean of the first evening hours of u* filtered night time NEE representing respiration. Consequently, we will amend the discussion according to the results of this analysis, although we do not believe that the outcome will change considerably.

3) We agree with reviewers that the current presentation of linear relationships between non gap-filled half-hourly day time or night time data and environmental drivers is inconclusive and that functional relationships of day time NEE with solar radiation, and night time NEE with soil/ air temperature would be a valuable addition to the manuscript. Thus, we will add such light response - and temperature response curves for each year to a

revised version of the manuscript. This will further allow an investigation of inter-annual differences in these functional relationships.
* * *
**Other concerns and issues:**
* * *
**Anonymous Referee #1**

**Main Concerns:**

**R1.1d:** It would seem that CO2 emissions from the diesel generator could appear as an erroneous CO2 source from the ecosystem. The text simply states that the generator is 'remote' but how remote can it be given concerns of power loss over long cables. How this concern is mitigated should be clarified.

Response: The remote area power system with the diesel generator is primarily there to power an automated soil GHG chamber system and directly located next to the tower (< 10 m) within the fenced component. Moreover, the generator is set to run only at night time for a couple of hours to recharge the battery bank. Given that we have observed only small night time fluxes we are confident that $CO_2$ emissions from the generator are not influencing our EC-tower measurements.

**R1.1e:** The explanation of footprint modeling and associated screening as well as screening based on wind direction so that "fluxes were constrained to the same forest type and dominant tree species" is inadequately described. Citing the Griebel et al. 2016 study is incomplete. The method should be explained more fully here.

Response: The footprint modelling was done by Griebel et al. (2016) in a separate study. For details on the method and further explanation the reader is directed to the Griebel et al. (2016) paper and we do not think it appropriate to be reproduced in this manuscript. However, we would rephrase the text in a revised version to the manuscript to:*"A footprint analysis by Griebel et al. (2016) using the parameterisation of flux footprint predictions of Kljun et al. (2004) showed that the distribution of fluxes were relatively homogeneous and that the whole footprint consisted of the same forest type and dominant tree species, and roughly uniform basal area. For further details see Griebel et al. (2016)."*

**R1.1f:** Measurement of soil moisture at 5 cm depth is unlikely to represent the soil water status relevant to 25 m tall trees. Measurement at a single point is also unlikely to be adequately representative given the typically large spatial variability of soil water content.

Response: We would like to clarify that the correct soil depth for soil moisture measurements was actually 10 cm soil depth and that soil moisture has been measured at and averaged over two micro-sites from 2011 onwards. We will amend the text in the instrumentation section accordingly. Soil moisture was also measured at 50 cm soil depth at one of the micro-sites. We will introduce this soil moisture data as well as model data from either BIOS2 or ACCESS-R in further data analysis to address concerns regarding depth and spatial representativeness.

**R1.1g:** Measurement of rainfall with a rain gage placed 1 m above the ground where trees are 25 m tall is sure to undersample the real rainfall rate above the canopy.

Response: At the time of installation of the rainfall gauge the surrounding area was relatively clear, i.e. no obstruction from trees, but we acknowledge that rainfall might be undersampled as rain falls seldom down straight. However, we currently have not used rainfall data from our study site for anything other than presenting annual rainfall amounts which are, as well as monthly rainfall data, in good agreement with data from the nearest Bureau of Meteorology rainfall station in Daylesford (11km N).

**R1.1i:** NEE separation was done four ways but then the results of only a single method were selected without adequate justification. It would seem more appropriate to present results from all four methods as a source of methodological uncertainty.

Response: We agree with the reviewer and will provide as supplementary materials a table with annual sums of ecosystem carbon fluxes derived from the four partitioning methods for comparison and a figure comparing the relative contribution of soil respiration data to the different outputs of ER.

**R1.1j:** A description of the method of NEE separation is needed. For example, what was the temporal window over which data were used to assess the relationship between ER (from selected Fc measurements) and environmental variables (principally temperature but also some others)?

Response: We will include a better description of the chosen partitioning method in section 2.3.3 and will also refer to the two methods papers in the same Special Issue: Beringer et al.

(2016) for DINGO and Isaac et al. (2016) for "OzFluxQC". In both papers detailed descriptions of various options of NEE partitioning methods can be found.

**R1.3a: L 296:** Comparison to these other forest types is valuable but the narrative is overly confident here. You cannot state that leaf longevity explains the differences in GPP rates between these various forests. The findings presented in this paper do not substantiate that supposition or conjecture.

Response: We will remove this statement.

**R1.3b: L304:** Delayed spring increase in ER relative to that for GPP may be partly due to soil respiration but it could still be partly linked to temperature control on plant respiration, no? Soil respiration is not shown in the present study and so this remains supposition.

Response: We agree with the review and propose to edit this section of the discussion accordingly.

**R1.3c: L335:** This site's very large carbon sink (NEE of around -1,000 g C m-2 y-1) is surprising and noteworthy. Could it be related to secondary recovery and the site's 25 year old stand age? What is the disturbance legacy for this site?

Response: The statement that the forest is a secondary regrowth forest refers to the heavy logging during the gold rush around 1850. General forest history includes harvesting and patchy occurrences of bushfires. Selective harvesting occurred until early 1970 when replaced by a more intensive shelterwood (two-stage clear-felling) system (Poynter, 2005). Since 2003 the Wombat State Forest has been under community forest management (Poynter, 2005) and harvesting has been strongly reduced. Forest management practices also include periodic low-intensity prescribed fires, and firewood collection in designated areas. Specific information on disturbances, i.e. harvests, wild fires or prescribed fires, for the study site are very limited. The study area was selectively harvested last in the early 1970s with the last bushfire on the outskirts of the study site recorded in 1982 and no recorded history of prescribed fires (DSE, 2012). We would like to clarify that the correct age of the forest within the study area could not clearly be determined and is rather of mixed age. The statement of ~25 years age rather referred to the last recorded disturbances. We will correct this information and will add more information of the site history in a revised version of the manuscript.

**R1.3d: L383:** It is not much of a finding to discover that seasonality is different in a dry, warm (winter free) temperate eucalypt site compared to temperate coniferous and deciduous forest sites with a strong seasonality in climate with sustained winter freezing. Furthermore, this study did not really show the difference in seasonality explicitly. Also, it is stated that this alleged difference in seasonality is due to the opportunistic response of eucalypt forests. This is not supported by any analysis and it is not even clear what is meant by "opportunistic response". A simpler explanation is that they in a different climate setting.

Response: This temperate evergreen forest was a near-continuous carbon sink with a distinct seasonality that was characterised by greater NEE in summer and lower NEE in winter. It will be important to differentiate this growth response from other, well characterised ecosystems and that is why comparisons with northern hemisphere ecosystems are made. The "opportunistic" nature of eucalypts refers to their broadleaf evergreen nature, which has been described in a number of papers before. More specifically it refers to their ability to respond to ideal growth conditions quickly, which is one of the main reasons for the absence of distinct growth rings in most eucalypts. We are happy to clarify this in more detail in the revised version of the manuscript and add more references.

**R1.3e: L 385:** This study did demonstrate "that seasonal and inter-annual variability in carbon uptake were not limited by temperature but predominantly driven by radiation". The study also did not demonstrate that "carbon loss from the forest was dominated and overall ecosystem carbon exchange dynamics were not water limited due to the high rainfall" and this sentence has a hanging statement (its first part about carbon loss).

Response: We will rephrase and finish the highlighted text sections accordingly.

**R1.3f: L 387:** Nothing presented in the paper quantitatively supports the statement that "temperate eucalypt forests represent a unique forest type and should be considered separately in future classifications of ecoystems...". To demonstrate this you would need to show that these forests have a different functional response to environmental conditions than other forest types. This has not been shown and would require a synthesis analysis, not simply data from a single site. It is entirely possible that if you were to control for site and time specific conditions of LAI, PAR, VPD, Tair, and soil moisture you might find that these forests behave similarly to others. This has not been tested in the present study.

Response: We agree and will delete this section.

**R1.3g: L 389:** Could drop the last sentence. It doesn't seem to add much of substance and I'd argue it does not really need to be said. It sounds a bit like a proposal statement or a sales pitch.

Response: We will delete this sentence.

**R1.4a:** Figure 2: it would be better to use a line chart for panel a rather than a bar chart.

Response: We are unclear what the reviewer is referring to. Fig 2 is a line chart for all continuous measurements and has a bar chart for the 7-day sums of precipitation data.

**R1.4b:** Figure 2: it would be helpful to show midday VPD either in addition to or instead of mean daily VPD.

Response: We will update Figure 2 to show mean midday VPD as 7-day running means in revised version of the manuscript.
* * *
**Anonymous Referee #2**

**Main concerns:**

**R2.3:** The u* threshold is unusually high at 0.56 to 0.69 m s-1. This is above most u*-thresholds reported in the literature (see for e.g. Papale et al. 2006 Biogeoscience Fig.1) Why is it so high? Greater details need to be given on how this threshold was determined. Is there really an inflection point around those values? In my experience, when the common methods determine such a high threshold, it is when they do not apply i.e. there is no inflection point because it is a site with advection or with some other abnormality making the u*-threshold determination not applicable.

Response: The u* threshold for our site is high, yes, but not unreasonably high when compared to other u* thresholds in the literature, e.g. Fig. 10 in Barr et al. (2013). We will provide in supplementary materials figures on the relationship between u* and night time NEE for each year clearly showing the inflection point, i.e. u* threshold. We will extend our current reference on details on how this threshold was determined, i.e. Barr et al. (2013), to both data processing papers for OzFlux in this Special Issue (Beringer et al., 2016; Isaac et al., 2016) that also briefly outline the adopted change point detection method by Barr et al. (2013).

**R2.4:** As a result of the very high u*-threshold, only a low number of actually measured data is kept for the three years (37%, 47%, 47%). This means that the outputted annual budgets depend mostly on the gap filling and thus, represent more modelled numbers rather than measurements. Sometimes, depending on the data quality, it can be better to use less ideal, measured data than modelled data. How different would the cumulative numbers be if for e.g. 60-70% of the measured data is kept?

Response: 1) We would like to clarify that the low percentage of available measured data is not only the result from the application of a u* threshold but also from the QA/QC process in which for example bad data due to rainfall has been discarded. We will provide in supplementary materials an overview table with percentages of yearly data loss due missing measurements, QA/QC process and u* filter application. 2) To increase the percentage of measured data would require to lower the u* threshold which would result in a bias of low NEE values. However, we will add an uncertainty analysis as described in McHugh et al. (2016) in this Special Issue which includes the effect of uncertainties in u* thresholds on annual NEE estimates by using the lower (5%) and upper (95%) confidence interval of the probability distribution of the mean u* threshold (Barr et al., 2013) and an uncertainty estimation of combined random and model error (Hollinger and Richardson, 2005; Keith et al., 2009). 3) We also would like to point out it is not unusual to have a high fraction of night time data removed through u* filtering and refer to Barr et al. (2013) who tested that even 90% of fractional night time data exclusion had a smaller impact on gap-filled NEE than the alternate option of underestimating the u* threshold.

**Minor comments:**
**Lines 51 to 57:** there are four "while" in those 7 lines, including two in one sentence. Please reduce the number of "while".
Response: We agree with the reviewer and will modify these three sentences to reduce the occurrence of "while".

**Line 66:** comma wrongly placed
Response: We will correct accordingly.

**Line 77:** comma missing between abundant and there
Response: We will correct accordingly.

**Line 105:** describe tree height better (how uniform and such) and add LAI in text.

Response: We will amend the text accordingly and please see response to Line 169 below.

**Line 114:** Is it really a mean of 114 years??

Response: Yes, it is the 114 year mean (1901 – 2014) of rainfall records from the nearest Bureau of Meteorology station to the study site.

**Lines 130-131:** Having one soil temperature, one soil moisture and one soil heat flux plate measurement is not sufficient to characterise soil due to its spatial variability. You need replicates horizontally and vertical profiles within the soil. On line 131, SWC is at 5 cm depth then on line 149 you talk about 10 cm depth. Please clarify the soil measurement depths.

Response: Soil temperature, soil moisture and soil heat flux have been measured at two micro sites and averaged over these two sites. The correct soil depth for SWC measurements is 10 cm soil depth. We will correct and clarify the text in the instrumentation section accordingly.

**Line 144:** There is an additional space before Contiuum.

Response: We will correct accordingly.

**Lines 145-146:** Greater details need to be given on the quality filtering. This is not clear. What range checks? What thresholds for spikes? What thresholds for outliers? etc. . .

Response: We will amend the text to clarify the process steps during the Ozflux QA/QC procedure and include a reference to the Isaac et al. (2016) paper in this Special Issue where the standard QA/QC filtering procedure is described in detail. This includes the standard application of range checks in plausible limits, spike detection, dependency checks and manual rejection of date ranges of all measured variables (covariances and meteorological variables) per month and year. When necessary, i.e. depending on site characteristics, these settings have been modified based on visual revision of the data during the QA/QC procedure.

**Line 169:** Is that how LAI was derived? Are there any ground measurements of LAI to validate the MODIS data?

Response: No, the LAI value (1.8) for this study site was obtained from another study (Moore, 2011) at the same study site (see Table 1) based on ground measurements (hemispherical images). The mean LAI of 1.8 has also been confirmed in a later study

(Griebel et al., 2016) and was based on ground measurements (hemispherical images). We will include this information in section 2.1 (site description) and citations in the discussion (section 4.3).

**Section 2.3.3:** it is confusing which method you used in the end for partitioning. Please give the equations used for partitioning and show a figure comparing the different approaches if you mention you compared different methods in text. Also, show a comparison with the soil respiration measurements that you mention on line 185.

Response: We agree with the reviewer and will make changes to the text to clarify that partitioning method (1) was used to estimate gross ecosystem carbon fluxes. We will provide as supplementary material a table with annual sums of ecosystem carbon fluxes derived from the four partitioning methods for comparison and a figure comparing the relative contribution of soil respiration data to the different outputs of ER.

**Line 301:** temperate evergreen coniferous forests (add evergreen). It is not because they are coniferous but rather because they are evergreen.

Response: We will correct accordingly.

**Figure 2(a)** is it NEE or –NEE (why is there a minus in the caption?) Same for Figure 3 and 4. Also, please describe what you display better, for e.g. in figure 3, what are the shaded lines at the back, the actual daily totals? Typically NEE is what is measured by EC, and sometimes people convert it to NEP=-NEE. In your case, you do not need the minus there because you display NEE.

Response: The reviewer is correct and we will remove the minus sign before NEE in all figure captions and throughout the text. We will also clarify and add additional description of figures 2, 3 and 4, i.e. the shaded lines in Figure 3 are daily totals (g C $m^{-2}$ $d^{-1}$) of ecosystem carbon fluxes, while bold lines represent 7-day running means of daily totals for better illustration.
* * *
**References:**

Barr, A. G., Richardson, A. D., Hollinger, D. Y., Papale, D., Arain, M. A., Black, T. A., Bohrer, G., Dragoni, D., Fischer, M. L., Gu, L., Law, B. E., Margolis, H. A., McCaughey, J. H., Munger, J. W., Oechel, W., and Schaeffer, K.: Use of change-point detection for friction-velocity threshold evaluation in eddy-covariance studies, Agricultural and Forest Meteorology, 171, 31-45, 2013.

Beringer, J., McHugh, I., Hutley, L. B., Isaac, P., and Kljun, N.: Dynamic INtegrated Gap-filling and partitioning for OzFlux (DINGO), Biogeosciences Discuss., 2016, 1-36, 2016.

DSE: http://nremap-sc.nre.vic.gov.au/MapShare.v2/imf.jsp?site=forestexplorer, last access: 02/08/2012 2012.

Finnigan, J.: The storage term in eddy flux calculations, Agricultural And Forest Meteorology, 136, 108-113, 2006.

Griebel, A., Bennett, L. T., Culvenor, D. S., Newnham, G. J., and Arndt, S. K.: Reliability and limitations of a novel terrestrial laser scanner for daily monitoring of forest canopy dynamics, Remote Sensing of Environment, 166, 205-213, 2015.

Griebel, A., Bennett, L. T., Metzen, D., Cleverly, J., Burba, G., and Arndt, S. K.: Effects of inhomogeneities within the flux footprint on the interpretation of seasonal, annual, and interannual ecosystem carbon exchange, Agricultural and Forest Meteorology, 221, 50-60, 2016.

Hollinger, D. and Richardson, A.: Uncertainty in eddy covariance measurements and its application to physiological models, Tree physiology, 25, 873-885, 2005.

Isaac, P., Cleverly, J., McHugh, I., van Gorsel, E., Ewenz, C., and Beringer, J.: OzFlux Data: Network integration from collection to curation, Biogeosciences Discuss., 2016, 1-41, 2016.

Kaimal, J. C. and Finnigan, J. J.: Atmospheric boundary layer flows: their structure and measurement, Oxford University Press, 1994.

Katul, G., Hsieh, C.-I., Bowling, D., Clark, K., Shurpali, N., Turnipseed, A., Albertson, J., Tu, K., Hollinger, D., and Evans, B.: Spatial variability of turbulent fluxes in the roughness sublayer of an even-aged pine forest, Boundary-Layer Meteorology, 93, 1-28, 1999.

Keith, H., Leuning, R., Jacobsen, K. L., Cleugh, H. A., van Gorsel, E., Raison, R. J., Medlyn, B. E., Winters, A., and Keitel, C.: Multiple measurements constrain estimates of net carbon exchange by a Eucalyptus forest, Agricultural and Forest Meteorology, 149, 535-558, 2009.

Kljun, N., Calanca, P., Rotachhi, M. W., and Schmid, H. P.: A simple parameterisation for flux footprint predictions, Boundary-Layer Meteorology, 112, 503-523, 2004.

Massman, W. J.: A simple method for estimating frequency response corrections for eddy covariance systems, Agricultural and Forest Meteorology, 104, 185-198, 2000.

McHugh, I. D., Beringer, J., Cunningham, S. C., Baker, P. J., Cavagnaro, T. R., Mac Nally, R., and Thompson, R. M.: Interactions between nocturnal turbulent flux, storage and advection at an 'ideal' eucalypt woodland site, Biogeosciences Discuss., 2016, 1-36, 2016.

Moore, C. E.: The surface water balance of the Wombat State Forest, Victoria: An estimation using Eddy Covariance and sap flow techniques, 2011.Honours Thesis, School of Geography and Envrionmental Science, Monash University, Melbourne, Australia, 104 pp., 2011.

Poynter, M.: Collaborative forest management in Victoria's Wombat State Forest — will it serve the interests of the wider community?, Australian Forestry, 68, 192-201, 2005.

van Gorsel, E., Leuning, R., Cleugh, H. A., Keith, H., and Suni, T.: Nocturnal carbon efflux: reconciliation of eddy covariance and chamber measurements using an alternative to the u(*)-threshold filtering technique, Tellus Series B-Chemical And Physical Meteorology, 59, 397-403, 2007.

---

## Author Response (AR2)

*Authors' response* to reviewer comments on "Net ecosystem carbon exchange of a dry temperate eucalypt forest" *by Hinko-Najera et al.* (bg-2016-192)

**Reviewer 1:**

The authors addressed well my concerns raised in the first review round with detailed analysis that improved the quality of the manuscript. I only have a few minor comments on this second version of the manuscript.

**Lines 16 to 18, Line 83:** There seems to be some confusion in the manuscript about the use of coniferous, deciduous, broadleaf and evergreen. On line 18, it would be clearer if instead of stating coniferous vs. deciduous, you would write broadleaf vs. coniferous OR evergreen vs. deciduous. Same throughout the manuscript.

Response: We have adjusted the used terms of "temperate coniferous" and "temperate deciduous" forests throughout the manuscript to "temperate evergreen coniferous" and "temperate deciduous broadleaved" forests as suggested by reviewer in line 360-361.

**Lines 49 to 52:** The entire manuscript only discusses about NEE so no need to introduce NEP. Same for the rest of the introduction. Please replace NEP by NEE.

Response: We agree with the reviewer and replaced the term NEP with the term NEE throughout the manuscript.

**Line 152:** A comma is missing after In February 2012.

Response: We corrected the text accordingly.

**Line 218:** Please rearrange equation 1 to remove NEP, i.e. NEE=ER-GPP

Response: We rearranged equation 1 accordingly.

**Lines 221-222:** the second part of the sentence linked to NEP can then be removed.

Response: We adjusted the sentence accordingly.

**Line 259:** either remove "this issue" or add in before

Response: We corrected the text accordingly.

**Line 278:** remove the extra period after SWC

Response: We corrected the text accordingly.

**Line 315:** "a increase" should be "an increase"

Response: We corrected the text accordingly.

**Line 346:** comma missing after Overall

Response: We corrected the text accordingly.

**Lines 360-361:** "Although daily maximum GPP rate at our forest site (14.7 g C m-2 d-1) were comparable with those from temperate evergreen coniferous forests (16.6-26.3 g C m-2 d-1), they were much lower than those reported for temperate deciduous broadleaved forests (22.4-31.0 g C m-2 d-1) during growing seasons (Falge et al., 2002)."

Response: We corrected the text accordingly.

**Line 370:** Remove link to unpublished literature, the reference to Fig. S2a is sufficient.

Response: We removed the citation of unpublished literature.

**Line 402:** There is a negative sign missing in front of 930.

Response: We corrected the text accordingly.

**Reviewer 2:**

The revised version has a core of methods, results, and interpretations that are acceptable for publication. The analysis of environmental drivers still needs to be improved. Below are suggestions for removal of a portion of the analysis, and replacement of this section by alternative methods that would provide more useful and powerful insights.

**1**) The authors have done a good job of revising their use of the eddy covariance technique, particularly the post-processing of data, to provide the best available dataset they can and in alignment with common practices. It is reassuring to see that results were so surprisingly robust to all of the changes that were made.

**2**) This lends needed confidence in all results up to the analyses of environmental drivers, and Figures 5, 6, and 7. Analyses of environmental drivers have also been significantly revised to remove circularity. However, the analyses still do not adequately address the second stated objective of the paper, to "identify the environmental controls of these CO2 ecosystem fluxes". Unfortunately, this portion of the paper is still not well designed and provides disappointingly limited insight.

Response: We agree with the reviewer and have extensively revised the analyses of environmental drivers in the manuscript to address our second objective (please see responses below). We also clarified the time scales in our objective as follows in section 1 of the manuscript: *"... 2) identify the environmental controls of these $CO_2$ ecosystem fluxes on seasonal and inter-annual time scales, ..."*

**3)** Analysis of Environmental Drivers: The random forest analysis attempts to estimate the relative importance of 4 environmental variables for determining midday-average NEE or early nighttime-average NEE for each month of the year and separately across the three years of study. Presumably there are thus at best only 30 or so observations in each bin on which the random forest is trained, which seems rather data scarce, and the reality is in fact far worse (<15) in many cases. Results in Fig 6 indicated that many of the months have fewer than 10 observations, and some have 2. How can a random forest possibly devise meaningful relationships regarding variable importance with so few observations? Apologies but this seems ludicrous. I recommend that analyses and results relating to Figures 5 and 6 be cut from the paper. (Note: For Figures 5 and 6, captions need to explain the numbers below the months on the x-axes, indicating the number of observations for each RF.) Figure 7 still has something to offer, but it could be replaced by something much better.

Response: We agree with the reviewer and have removed Figures 5 and 6 as well as the Random Forest analyses all together (please see point 4) below). We acknowledge that during some time periods (i.e. particularly during winter) data observations were not sufficient to analyse individual months per year. Moreover, we did not identify inter-annual differences in environmental controls of NEE and hence, we have pooled data across years for each month for seasonal analyses of environmental controls of NEE. Please see our response below regarding our revised analysis of environmental controls in point 4).

**4)** As suggested by Reviewer 2 in the prior review, the RF approach is rather indirect for gleaning insights into underlying processes. The paper does not provide a clear diagnosis of environmental controls on CO2 fluxes, but it could. Alternative, more fruitful approaches are available. As stated by R2.6, the random forest analysis could be replaced by analysis of functional relationships, conditionally sampling data to reveal light response parameters (e.g. NEE at light saturation and with low to modest VPD and for low versus high soil moisture), and similarly for VPD response, soil moisture response, and temperature response. The author's comment in open review ignored this suggestion / critique by R2 altogether. Numerous studies have shown how this can be done with eddy covariance data but this study does not follow those leads for some reason.

Response: We followed the advice from the reviewer and analysed functional relationships between NEE (day time and night time) and selected environmental drivers. We would like to point out that the results of this analysis, particularly the seasonal variability in the environmental controls for day time and nigh time NEE, were in agreement with those from the Random Forest analysis and did not change the overall outcome of environmental controls on NEE in this forest during the presented study period. However, we entirely replaced the Random Forest analysis with the new analysis of environmental drivers based on functional relationships. The analysis is outlined below and we have revised the relevant sections (2.4, 3.3, 4.2) in the manuscript accordingly.

As previously outlined in the manuscript we used daily means of quality controlled half-hourly non gap filled midday NEE (hours 11:00 – 13:00) for day time NEE and daily means of half-hourly quality controlled non gap filled and u* filtered night time NEE. We would like to note that while daily means have been used, no change in results of analyses were found using daily means or half-hourly observations of selected data as the above outlined data selection already excludes any diurnal influence on the analysis of seasonal environmental controls.

**4.1) In regard to day time NEE:** We analysed the dependency of day time NEE on incoming solar radiation (Fsd) using a rectangular hyperbolic light response curve (LRC) or *Michaelis-Menten* equation (Carrara et al., 2004; Falge et al., 2001; Flanagan et al., 2002; Lasslop et al., 2010; Michaelis and Menten, 1913). We would like to note that other published variations of a LRCs were tested but they either performed inferior to the above mentioned LRC or resulted in arbitrary parameter estimates: modifications of the rectangular hyperbolic curve (Falge et al., 2001), a non-rectangular hyperbolic curve (Gilmanov et al., 2007; 2003) and a logistic sigmoid function (Eugster et al., 2010; Wolf et al., 2011). As midday NEE represents the peak of photosynthetic activity we found that the respiration parameter was marginal and insignificant for the fit of the function or the parameter of maximum NEE (i.e. uptake rate of the canopy) at light saturation. Therefore we removed the respiration parameter to improve significance of curve fit and slope of LRC (i.e. the canopy light utilization efficiency) (Flanagan et al., 2002).

Residuals of the LRC were then used to analyse the dependency of NEE on either air temperature (Ta) or vapour pressure deficit (VPD) given the dependency of VPD on Ta and thus strong auto correlation (Carrara et al., 2004; Chen et al., 2002). Relationships between residuals of the LRC and Ta or VPD were tested with linear and non linear regressions, i.e. exponential temperature sensitivity functions according to Lloyd and Taylor (1994) for Ta and a logarithmic power model according to Chen et al. (2002) for VPD. However, for both, Ta and VPD, linear relationships resulted in the best fits whereas non linear regressions consistently resulted in arbitrary or insignificant parameter estimates.

A potential influence of soil water content (SWC) on day time NEE was tested with linear regressions between SWC and residuals of LRC and $2^{nd}$ residuals from the linear relationships between LRC residuals and Ta as temperature and soil moisture are often negatively correlated in this forest ecosystem (Hinko-Najera et al., 2015).

In addition we analysed LRCs with data divided into various Ta bins, VPD bins and SWC bins (see Table R3).

Results of the LRC fits and linear fits with T and VPD per year and seasons are given in Table R1 and overall fit is displayed in Figure R1.a,b and c. Overall Fsd could explain 25% of the temporal variability in midday NEE which did not considerable vary between observation years. Similarly

both T and VPD explained about 18% or 23% of the overall temporal variability in midday NEE and again no considerable inter-annual differences were determined. However, a clear distinct seasonal pattern was shown in the dependencies of midday NEE on Fsd, T and VPD when coefficients of determinations are plotted for each month (Figure R2.a). While Fsd was the dominant environmental driver during mid/ late autumn and winter months (36 – 47%, mean = 42%), Ta and VPD were the main controlling environmental variables during spring, summer and early autumn months (23 – 56%, mean = 40% for Ta and 15 – 48%, mean = 31% for VPD). LRCs fitted for various Ta bins and VPD bins (Table R3) show a strong decrease in the net carbon uptake at temperatures above 20°C or VPD values above 1.2 kPa. A clear differentiation between Ta and VPD was very difficult because of the strong correlation between VPD and Ta. While overall and annual variability in residuals of midday NEE were marginally better explained by VPD than Ta, variability of midday NEE residuals from spring to early autumn correlated stronger with changes in Ta than VPD. Considering the high rainfall during the observation years including summer months it is likely that midday NEE is limited by higher temperatures (i.e. increasing ER) than high VPD or water stress on photosynthetic activity.

In accordance with a greater temperature effect than effect of water stress on midday NEE during spring and summer months is the absence of a clear influence of SWC on residuals of NEE (Table R1.c, R3)

**4.2) In regard to night time NEE:**

The dependency of night time NEE on temperature was analysed using an Arrhenius-type model function (LT) (Lloyd and Taylor, 1994). Relationships were analysed with either Ta or soil temperature at 10 cm soil depth and consistently best fits were achieved with air temperature for every subset of data. Residuals of LT were then used to analyse the dependency of night time NEE on SWC with linear regressions. We also unsuccessfully tested non-linear regressions (hyperbolic function) between LT residuals and SWC. In addition we analysed LTs with data divided into various SWC bins (Table R2).

Results of LT and linear fits with SWC for various subsets of data are given in Table R2 and overall temperature sensitivity of night time NEE is displayed in Figure R1.d. Overall 36% of the temporal variability in u* filtered night time NEE was explained by temperature which varied from 30% in 2012 to 31% in 2010 and 49% in 2011. On seasonal time scales the dependence of night time NEE on Ta strongly varied being greatest during spring (39 – 44%, mean = 42%) followed by summer months (31 – 45%, mean = 38%) and lowest during autumn months (15 – 30%, mean = 24%) (Figure R2.b). No significant relationships could be determined during winter months where greater data gaps occurred compared to other months. Neither LT fitted for SWC bins (Table R2) nor linear

relationships between SWC and LT residuals (not shown) showed an influence of SWC on night time NEE with the exception of the winter month July where night time NEE decreased with increasing SWC ($R^2 = 0.26$ ***). This would indicate a limitation of respiration due to high water content, however such an indication is precautious as data availability was lowest (<50%) during July.

Table R1: Parameters, standard errors and/or coefficient of determination ($R^2$) of (a) the rectangular hyperbolic light response curve (LRC) between daily means of midday NEE and incoming radiation (Fsd), (b) linear fits between residuals of LRC and air temperature (Ta) or vapour pressure deficit (VPD) and (c) linear fits between 2nd residuals (b) with Ta and soil water content (SWC) for subsets of data, α: initial slope of LRC and canopy light utilization efficiency ($\mu$mol $CO_2$ $J^{-1}$), β: maximum NEE (i.e. uptake rate of the canopy) at light saturation ($\mu$mol $CO_2$ $m^{-2}$ $s^{-1}$); significance level: *** <0.001, ** <0.01, * <0.5, ns: not significant

| data subset | (a) Fsd | | | (b) T | VPD | (c) SWC | nr |
|---|---|---|---|---|---|---|---|
| | α (se) | β (se) | $R^2$ | $R^2$ | $R^2$ | $R^2$ | |
| All data | -0.14 0.01 *** | -21.8 0.0 *** | 0.25 | 0.18 *** | 0.23 *** | 0.05 *** | 792 |
| 2010 | -0.11 0.02 *** | -22.5 0.0 *** | 0.28 | 0.19 *** | 0.25 *** | 0.05 *** | 213 |
| 2011 | -0.13 0.02 *** | -21.9 0.0 *** | 0.27 | 0.21 *** | 0.21 *** | 0.10 *** | 292 |
| 2012 | -0.17 0.02 *** | -21.3 0.0 *** | 0.22 | 0.15 *** | 0.24 *** | 0.04 *** | 287 |

Table R2: Parameters, standard errors and/or coefficient of determination ($R^2$) of (a) the temperature response function after Lloyd and Taylor (1994) between daily means of u* filtered night time NEE and air temperature (Ta) for subsets of data, $R_{ref}$: basal respiration rate at 10°C ($\mu$mol $CO_2$ $J^{-1}$), E: activation energy related parameter; significance level: *** <0.001, ** <0.01, * <0.5, ns: not significant

| LT | $R_{ref}$ (se) | E (se) | $R^2$ | nr |
|---|---|---|---|---|
| All data | 3.0 0.1 *** | 310 16 *** | 0.36 | 694 |
| 2010 | 3.6 0.2 *** | 215 23 *** | 0.31 | 214 |
| 2011 | 2.8 0.2 *** | 405 27 *** | 0.49 | 234 |
| 2012 | 2.8 0.2 *** | 306 31 *** | 0.30 | 246 |
| SWC bins | | | | |
| 0.1-<0.15 | 2.1 0.5 *** | 349 94 *** | 0.34 | 37 |
| 0.15-<0.20 | 3.0 0.3 *** | 285 44 *** | 0.27 | 138 |
| 0.20-<0.25 | 3.1 0.2 *** | 339 34 *** | 0.38 | 146 |
| 0.25-<0.30 | 3.4 0.2 *** | 337 30 *** | 0.38 | 200 |
| >0.30 | 3.3 0.2 *** | 527 40 *** | 0.42 | 173 |

[Figure]

Figure R1: (a-c) Relationship between daily means of midday NEE and (a) incoming radiation (Fsd) in a rectangular hyperbolic light response curve (LRC), linear fits between residuals of LRC and (b) air temperature (Ta) or (c) vapour pressure deficit (VPD) and (d) relationship between daily means of u* filtered night time NEE and air temperature as temperature (Ta) response function after Lloyd and Taylor (1994), $R^2$ are given in Table R1 and R2

[Figure]

Figure R2: Seasonal variability in environmental controls of day time and night time NEE; coefficients of determination ($R^2$) of (a) the rectangular hyperbolic light response curve (LRC) between daily means of midday NEE and incoming radiation (Fsd, black lines and circles), linear fits between residuals of LRC and air temperature (Ta, red lines and triangles) or vapour pressure deficit (VPD, blue lines and diamonds) and (b) the temperature response function after Lloyd and Taylor (1994) between daily means of u* filtered night time NEE and air temperature (Ta, black lines and circles) per month pooled over three years; open symbols indicate non significant $R^2$; nr per month for (a) from Jan to Dec: 62, 74, 77, 76, 58, 42, 56, 60, 74, 75, 66, 72; nr per month for (b) from Jan to Dec: 55, 69, 63, 66, 52, 45, 41, 57, 63, 63, 52; 68

Table R3: Parameters, standard errors and/or coefficient of determination ($R^2$) of the rectangular hyperbolic light response curve (LRC) between daily means of midday NEE and incoming radiation (Fsd) for various Ta bins, VPD bins and SWC bins; $\alpha$: initial slope of LRC and canopy light utilization efficiency ($\mu$mol $CO_2$ $J^{-1}$), $\beta$: maximum NEE (i.e. uptake rate of the canopy) at light saturation ($\mu$mol $CO_2$ $m^{-2}$ $s^{-1}$); significance level: *** <0.001, ns: not significant

| LRC | $\alpha$ (se) | | | $\beta$ (se) | | | $R^2$ | nr |
|---|---|---|---|---|---|---|---|---|
| **Ta bins** | | | | | | | | |
| **<8** | -0.14 | 0.02 | *** | -23.6 | 0.0 | *** | **0.39** | 133 |
| **8-12** | -0.09 | 0.01 | *** | -29.3 | 0.0 | *** | **0.55** | 205 |
| **12-16** | -0.07 | 0.01 | *** | -30.4 | 0.0 | *** | **0.50** | 172 |
| **16-20** | -0.07 | 0.01 | *** | -27.9 | 0.0 | *** | **0.37** | 124 |
| **20-24** | -0.08 | 0.02 | *** | -20.7 | 0.0 | *** | **0.19** | 97 |
| **24-28** | -0.06 | 0.03 | ns | -20.5 | 0.1 | *** | 0.13 | 49 |
| **VPD bins** | | | | | | | | |
| **<0.4** | -0.10 | 0.01 | *** | -28.7 | 0.0 | *** | **0.46** | 324 |
| **0.4-0.8** | -0.07 | 0.01 | *** | -30.5 | 0.0 | *** | **0.44** | 232 |
| **0.8-1.2** | -0.05 | 0.01 | *** | -32.9 | 0.0 | *** | **0.47** | 103 |
| **1.2-1.6** | -0.62 | 1.39 | ns | -16.7 | 0.7 | *** | 0.00 | 68 |
| **1.6-2.0** | -0.16 | 0.25 | ns | -16.3 | 0.5 | *** | 0.02 | 25 |
| **2.0-2.4** | -0.06 | 0.05 | ns | -18.7 | 0.2 | *** | 0.09 | 26 |
| **2.4-2.8** | -0.05 | 0.03 | ns | -15.1 | 0.2 | *** | 0.47 | 7 |
| **2.8-3.2** | - | | | - | | | - | |
| **>3.2** | -0.08 | 0.27 | ns | -12.7 | 0.8 | *** | 0.05 | 4 |
| **SWC bins** | | | | | | | | |
| **<0.22** | -0.11 | 0.02 | *** | -22.35 | 0.0 | *** | **0.18** | 258 |
| **>=0.22** | -0.14 | 0.01 | *** | -21.88 | 0.0 | *** | **0.29** | 534 |

**5)** Furthermore, the existing analysis of which variables had greatest importance for driving within-month or within-year variability leaves us wondering what the relationships look like. Is there a positive or negative relationship between sunlight and midday NEE? How strong is the relationship? Same for all of the other potential drivers.

Response: We agree with the reviewer and we have addressed these questions in our response to point 4) above.

**6)** The methods description does not indicate how data were binned for the random forest analysis of within-year variability.

Response: This clarification is no longer relevant as we replaced the Random Forest analysis with the above described analysis of environmental drivers with functional relationships.

**References:**

Carrara, A., Janssens, I. A., Yuste, J. C., and Ceulemans, R.: Seasonal changes in photosynthesis, respiration and NEE of a mixed temperate forest, Agricultural And Forest Meteorology, 126, 15-31, 2004.

Chen, J., Falk, M., Euskirchen, E., Paw U, K. T., Suchanek, T. H., Ustin, S. L., Bond, B. J., Brosofske, K. D., Phillips, N., and Bi, R.: Biophysical controls of carbon flows in three successional Douglas-fir stands based on eddy-covariance measurements, Tree Physiology, 22, 169-177, 2002.

Eugster, W., Moffat, A. M., Ceschia, E., Aubinet, M., Ammann, C., Osborne, B., Davis, P. A., Smith, P., Jacobs, C., Moors, E., Le Dantec, V., Beziat, P., Saunders, M., Jans, W., Grunwald, T., Rebmann, C., Kutsch, W. L., Czerny, R., Janous, D., Moureaux, C., Dufranne, D., Carrara, A., Magliulo, V., Di Tommasi, P., Olesen, J. E., Schelde, K., Olioso, A., Bernhofer, C., Cellier, P., Larmanou, E., Loubet, B., Wattenbach, M., Marloie, O., Sanz, M. J., Sogaard, H., and Buchmann, N.: Management effects on European cropland respiration, Agriculture Ecosystems & Environment, 139, 346-362, 2010.

Falge, E., Baldocchi, D., Olson, R., Anthoni, P., Aubinet, M., Bernhofer, C., Burba, G., Ceulemans, R., Clement, R., Dolman, H., Granier, A., Gross, P., Grunwald, T., Hollinger, D., Jensen, N. O., Katul, G., Keronen, P., Kowalski, A., Lai, C. T., Law, B. E., Meyers, T., Moncrieff, H., Moors, E., Munger, J. W., Pilegaard, K., Rannik, U., Rebmann, C., Suyker, A., Tenhunen, J., Tu, K., Verma, S., Vesala, T., Wilson, K., and Wofsy, S.: Gap filling strategies for defensible annual sums of net ecosystem exchange, Agricultural And Forest Meteorology, 107, 43-69, 2001.

Flanagan, L. B., Wever, L. A., and Carlson, P. J.: Seasonal and interannual variation in carbon dioxide exchange and carbon balance in a northern temperate grassland, Global Change Biology, 8, 599-615, 2002.

Gilmanov, T. G., Soussana, J. E., Aires, L., Allard, V., Ammann, C., Balzarolo, M., Barcza, Z., Bernhofer, C., Campbell, C. L., Cernusca, A., Cescatti, A., Clifton-Brown, J., Dirks, B. O. M., Dore, S., Eugster, W., Fuhrer, J., Gimeno, C., Gruenwald, T., Haszpra, L., Hensen, A., Ibrom, A., Jacobs, A. F. G., Jones, M. B., Lanigan, G., Laurila, T., Lohila, A., Manca, G., Marcolla, B., Nagy, Z., Pilegaard, K., Pinter, K., Pio, C., Raschi, A., Rogiers, N., Sanz, M. J., Stefani, P., Sutton, M., Tuba, Z., Valentini, R., Williams, M. L., and Wohlfahrt, G.: Partitioning European grassland net ecosystem CO2 exchange into gross primary productivity and ecosystem respiration using light response function analysis, Agriculture Ecosystems & Environment, 121, 93-120, 2007.

Gilmanov, T. G., Verma, S. B., Sims, P. L., Meyers, T. P., Bradford, J. A., Burba, G. G., and Suyker, A. E.: Gross primary production and light response parameters of four Southern Plains ecosystems estimated using long-term CO2-flux tower measurements, Global Biogeochemical Cycles, 17, 16, 2003.

Hinko-Najera, N., Fest, B., Livesley, S. J., and Arndt, S. K.: Reduced throughfall decreases autotrophic respiration, but not heterotrophic respiration in a dry temperate broadleaved evergreen forest, Agricultural and Forest Meteorology, 200, 66-77, 2015.

Lasslop, G., Reichstein, M., Papale, D., Richardson, A. D., Arneth, A., Barr, A., Stoy, P., and Wohlfahrt, G.: Separation of net ecosystem exchange into assimilation and respiration using a light response curve approach: critical issues and global evaluation, Global Change Biology, 16, 187-208, 2010.

Lloyd, J. and Taylor, J. A.: On the temperature dependence of soil respiration, Functional Ecology, 8, 315-323, 1994.

Michaelis, L. and Menten, M. L.: Die kinetik der invertinwirkung, Biochem. z, 49, 352, 1913.

Wolf, S., Eugster, W., Potvin, C., Turner, B. L., and Buchmann, N.: Carbon sequestration potential of tropical pasture compared with afforestation in Panama, Global Change Biology, 17, 2763-2780, 2011.